# Polo-like kinase acts as a molecular timer that safeguards the asymmetric fate of spindle microtubule-organizing centers

**Laura Matellán, Javier Manzano-López, Fernando Monje-Casas***

Centro Andaluz de Biología Molecular y Medicina Regenerativa (CABIMER) / Spanish National Research Council (CSIC) - University of Seville - University Pablo de Olavide, Sevilla, Spain

**Abstract** The microtubules that form the mitotic spindle originate from microtubule-organizing centers (MTOCs) located at either pole. After duplication, spindle MTOCs can be differentially inherited during asymmetric cell division in organisms ranging from yeast to humans. Problems with establishing predetermined spindle MTOC inheritance patterns during stem cell division have been associated with accelerated cellular aging and the development of both cancer and neurodegenerative disorders. Here, we expand the repertoire of functions Polo-like kinase family members fulfill in regulating pivotal cell cycle processes. We demonstrate that the Plk1 homolog Cdc5 acts as a molecular timer that facilitates the timely and sequential recruitment of two key determinants of spindle MTOCs distribution, that is the γ-tubulin complex receptor Spc72 and the protein Kar9, and establishes the fate of these structures, safeguarding their asymmetric inheritance during *Saccharomyces cerevisiae* mitosis.

## Introduction

To maintain correct ploidy through mitosis, cells have developed an elaborate molecular machinery that facilitates chromosomal segregation and surveillance mechanisms that preserve DNA integrity and ensure even distribution of the duplicated genome during the process. A critical component of this system is the mitotic spindle, a bipolar array of microtubules that originate from microtubule-organizing centers (MTOCs) located at either pole. These MTOCs, known as centrosomes in animal cells or spindle pole bodies (SPBs) in budding yeast (*Conduit et al., 2015*; *Ito and Bettencourt-Dias, 2018*; *Kilmartin, 2014*), nucleate interpolar microtubules that provide stability to the spindle, kinetochore microtubules that anchor the chromosomes to enable their segregation, and astral or cytoplasmic microtubules that position the spindle relative to the cytokinesis plane (*Prosser and Pelletier, 2017*; *Winey and Bloom, 2012*). After duplication early in the cell cycle, spindle MTOCs exhibit inherent asymmetry, with pre-existent ('old') and newly generated ('new') MTOCs differing in terms of composition, structure, and age (*Nigg and Stearns, 2011*; *Pelletier and Yamashita, 2012*). The old and new spindle MTOCs can be differentially distributed between the mother and daughter cells during certain asymmetric divisions (*Lengefeld and Barral, 2018*; *Pelletier and Yamashita, 2012*; *Reina and Gonzalez, 2014*). Originally described in the budding yeast *Saccharomyces cerevisiae* (*Pereira et al., 2001*), this phenomenon was later also documented in cells from other organisms, including humans (*Izumi and Kaneko, 2012*; *Pelletier and Yamashita, 2012*; *Reina and Gonzalez, 2014*). We have recently demonstrated that the asymmetric SPB inheritance pattern is essential for maintaining the full replicative lifespan of budding yeast cells (*Manzano-López et al., 2019*).

The precise mechanisms that orchestrate the differential distribution of old and new spindle MTOCs during asymmetric cell divisions are still not completely understood. However, many

***For correspondence:**
fernando.monje@cabimer.es

**Competing interests:** The authors declare that no competing interests exist.

proteins involved in this process are evolutionarily conserved; an illustrative example is the CDK5RAP2 family of γ-tubulin complex receptors (γ-TuCRs). Spc72, a member of this family, asymmetrically localizes to the SPB that enters the daughter cell during budding yeast division, and is required for establishing the differential SPB inheritance pattern during mitosis (*Juanes et al., 2013*). Analogously, centrosomin (*Cnn*), the *Drosophila* CDK5RAP2 homolog, is required for asymmetric centrosome inheritance in germline stem cells (GSCs) and neuroblasts (*Conduit and Raff, 2010*; *Yamashita et al., 2007*). Centrosomes are also differentially inherited during the division of mouse radial glia progenitors and human neuroblastoma cells (*Conduit and Raff, 2010*; *Izumi and Kaneko, 2012*; *Rebollo et al., 2007*; *Wang et al., 2009*). Based on the importance of neural progenitor asymmetric division for generating the different cells that compose the brain and central nervous system, these observations suggest a possible role of the non-random distribution of centrosomes during brain development. CDK5RAP2 is essential for determining cell fate during the division of apical progenitors in mouse brain neuroepithelium (*Buchman et al., 2010*; *Lizarraga et al., 2010*). Moreover, several human brain diseases arise from problems with spindle positioning that perturb neural progenitor asymmetric division; one such example is autosomal recessive primary microcephaly (MCPH) (*Barbelanne and Tsang, 2014*; *Faheem et al., 2015*; *Lancaster and Knoblich, 2012*). Most genes linked to MCPH encode proteins required for proper centrosome function and spindle orientation (*Barbelanne and Tsang, 2014*; *Faheem et al., 2015*). Based on the evidence that links differential spindle MTOC distribution with the pathways that control cell differentiation and the establishment of the replicative lifespan, it is of utmost importance to find new factors that act in this process. Subsequently, it could help explain how defects during asymmetric stem cell division could be at the origin of age-related diseases in humans, such as neurodegenerative disorders or cancer.

Initial evidence in *Drosophila* support the premise that Polo-like kinases, another highly conserved protein family (*Archambault and Glover, 2009*), also contribute to conferring a differential identity to both centrosomes during asymmetric mitoses. In *Drosophila* neuroblasts, POLO is important for controlling the unequal mother–daughter behavior of centrioles (*Januschke et al., 2013*). Cdc5, the only Polo-like kinase in budding yeast, localizes to the SPBs and has an important role during SPB duplication and maturation (*Elserafy et al., 2014*; *Ratsima et al., 2016*; *Song et al., 2000*). To better understand the precise mechanisms by which Polo-like kinases might facilitate asymmetric spindle MTOC distribution, we evaluated the possible role of Cdc5 during the establishment of the SPB inheritance pattern in *S. cerevisiae*. We demonstrate that Cdc5 safeguards SPB differential distribution during mitosis by acting as a molecular timer that ensures timely and sequential recruitment of Spc72 and Kar9, two key determinants of SPB inheritance, to the SPBs. Our results shed light on the complex regulatory network that cells form to enable the generation of non-random inheritance patterns of spindle-associated MTOCs. Our findings will aid better understanding of the pathways that control cell fate determination and aging.

## Results

### Asymmetric SPB inheritance in *S. cerevisiae* requires Cdc5 activity

During budding yeast division, SPBs are differentially distributed in anaphase so that the daughter cell preferentially inherits the old SPB, while the mother cell retains the new SPB (*Pereira et al., 2001*). SPB age can be discriminated by tagging the constitutive SPB component Spc42 with red fluorescent protein (RFP) (*Pereira et al., 2001*). The slow-folding properties of RFP and the mostly conservative nature of SPB duplication ensured that the new SPB, which mostly incorporated fluorescently inactive Spc42-RFP, displayed a much weaker fluorescent signal than the old SPB (*Figure 1A, B*). To evaluate whether Cdc5 has a role in regulating asymmetric SPB inheritance, we analyzed it in cells expressing the *cdc5-as1* allele, which encodes a mutant kinase that can be conditionally inactivated with the inhibitory ATP analogue CMK-C1 (*Snead et al., 2007*). Cdc5 inhibition in an asynchronous culture of exponentially growing cells revealed randomized SPB segregation (*Figure 1B*). Similar results were obtained when SPB inheritance was evaluated using the dsRed-tagged integral SPB component Spc110, which folds slightly faster into a fluorescently active molecule and facilitates discrimination of old from new SPBs, while at the same time allowing visualization of both spindle MTOCs (*Moore et al., 2006*; *Figure 1—figure supplement 1A,B*). Randomization of SPB fate was

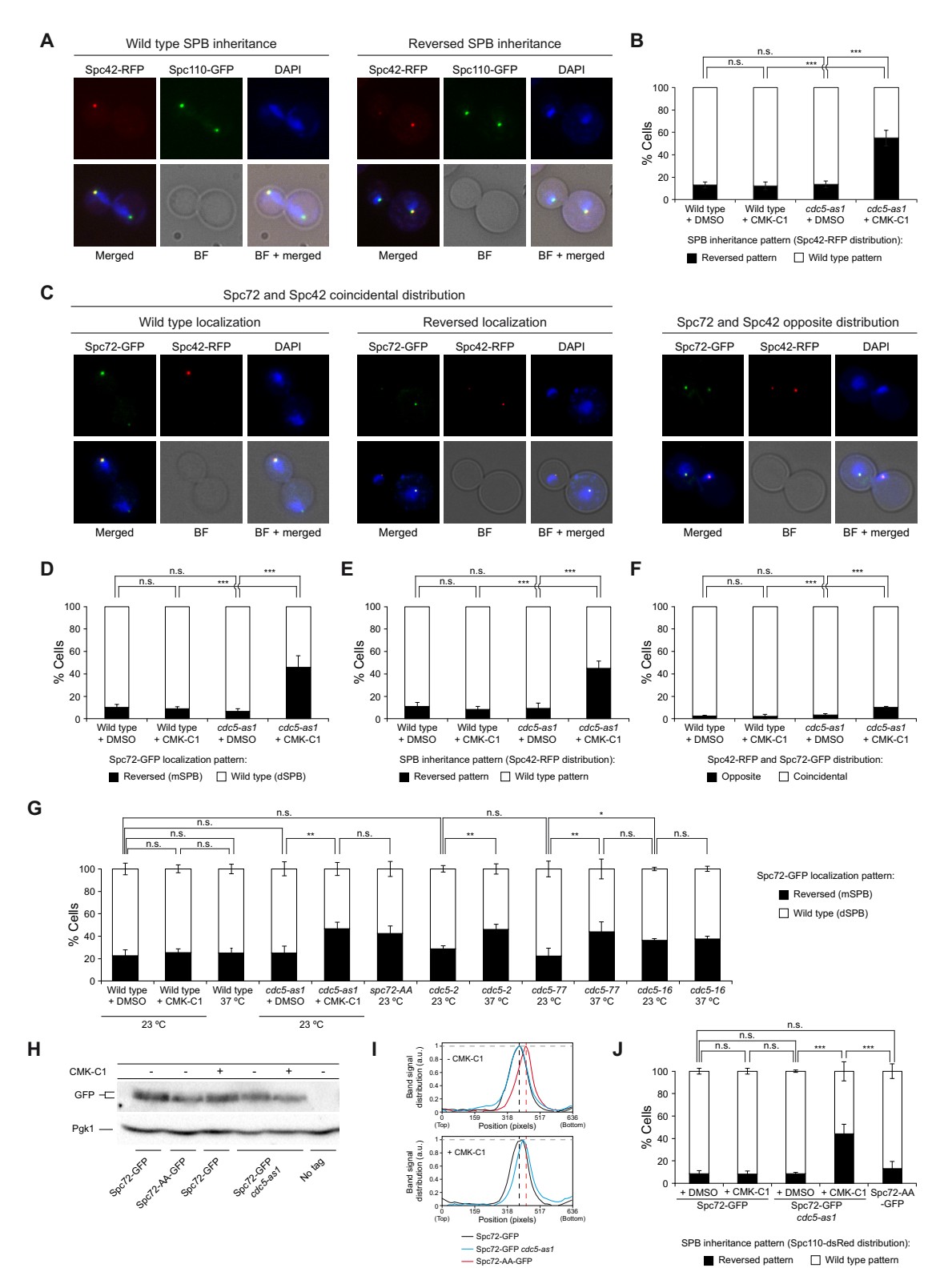

**Figure 1.** Cdc5 activity is required for asymmetric SPB inheritance and Spc72 distribution. (A, B) SPB inheritance in wild-type (F2071) and *cdc5-as1* (F2741) cells expressing Spc42-RFP and Spc110-GFP. (A) Representative images of Spc42-RFP (red) and Spc110-GFP (green) localization in cells exhibiting wild-type and reversed (old SPB retained in the mother cell) SPB distribution patterns. Also shown are nuclear morphology (DAPI, blue), bright-field (BF), and merged images. (B) The percentage of cells displaying wild-type (white bars) or reversed (black bars) SPB inheritance, based on

*Figure 1 continued on next page*

*Figure 1 continued*

Spc42-RFP distribution. (C–F) Spc72-GFP distribution and SPB inheritance in wild-type (F3712) and *cdc5-as1* (F3707) cells also expressing Spc42-RFP. (C) Representative images showing coincidental or opposite Spc72-GFP (green) and Spc42-RFP (red) distribution, distinguishing in the former between cells exhibiting wild-type (main accumulation in the bud) and reversed (preferential retention in the mother cell) Spc72 localization. Also shown are nuclear morphology (DAPI, blue), BF, and merged images. (D–F) The percentages of cells displaying wild-type (white bars) or reversed (black bars) Spc72-GFP localization (D), wild-type (white bars) or reversed (black bars) SPB inheritance (based on Spc42-RFP distribution) (E), and coincidental (white bars) or opposite (black bars) distribution of both tagged proteins (F). (G) Spc72-GFP distribution in wild-type (F3696), *cdc5-as1* (F3699), *cdc5-2* (F4439), *cdc5-77* (F4705), and *cdc5-16* (F4699) cells, and localization of Spc72-AA-GFP mutant protein in a wild-type background (F4606). (H, I) Western blot analyses in asynchronous cultures of wild-type (F3696) or *cdc5-as1* (F3699) cells expressing Spc72-GFP, and of wild-type cells expressing the Spc72-AA-GFP mutant (F4606) or untagged Spc72 (F496, No tag). (H) The experiment was performed thrice (n = 3); shown is a representative image displaying the levels of GFP-tagged protein. Pgk1 levels were used as the loading control. (I) Intensity plot profile of the lanes from the blot shown in (H). (J) SPB inheritance in wild-type (F3702) and *cdc5-as1* (F3705) cells expressing Spc72-GFP, and in wild-type cells expressing Spc72-AA-GFP (F4608), all simultaneously carrying Spc110-dsRed. Shown is the percentage of wild-type (white bars) or reversed (black bars) SPB inheritance, based on Spc110-dsRed distribution. (B, D, E, F, G, J) Final data are the average of three experiments (n = 3; 100 cells each). Error bars represent the SD. Statistical significance according to the Newman-Keuls multiple comparison test is indicated.

The online version of this article includes the following figure supplement(s) for figure 1:

**Figure supplement 1.** SPB inheritance and Spc72 post-translational modification in cells lacking Cdc5 activity.

also observed when Cdc5 activity was inhibited by shifting cells carrying the *cdc5-2* or *cdc5-77* thermosensitive alleles at a restrictive temperature (*Hu et al., 2001*; *Ratsima et al., 2016*). The results demonstrate that this phenotype is not a specific defect associated with *cdc5-as1* (*Figure 1—figure supplement 1C*).

## Spc72 and Kar9 distribution depends on Cdc5 kinase activity

The asymmetric recruitment of Spc72 to the SPBs represents the most upstream event that ensures the differential segregation of budding yeast SPBs (*Juanes et al., 2013*). Spc72 interacts and is phosphorylated by Cdc5 (*Maekawa et al., 2007*; *Snead et al., 2007*). In accordance with those data, the Spc72-13Myc fusion protein was increasingly modified post-translationally, peaking in anaphase as the cells synchronously progressed into mitosis, whereas inhibiting Polo-like kinase activity in *cdc5-as1* cells markedly impaired the modification (*Figure 1—figure supplement 1D,E*). Therefore, Spc72 represents an ideal candidate for mediating Cdc5 function during the establishment of the differential SPB inheritance pattern in budding yeast.

To evaluate whether Cdc5 could facilitate asymmetric SPB inheritance by regulating Spc72 distribution to these structures, we analyzed the localization of GFP (green fluorescent protein)-tagged Spc72 in cells expressing *cdc5-as1* (*Figure 1C–F*). Despite Spc72-GFP being observed on both SPBs, it localized asymmetrically from the onset of SPB separation until full elongation of the mitotic spindle in late anaphase; the cells displayed a stronger fluorescent signal at the daughter cell–destined SPB (dSPB, *Figure 1C,D*; *Juanes et al., 2013*). Inhibiting Cdc5 activity in cells from asynchronous culture led to randomization of both Spc72 distribution and SPB fate (*Figure 1C–E*). Again ruling out an allele-specific defect, Spc72-GFP distribution was also randomized in *cdc5-2* and *cdc5-77* cells at the restrictive temperature (*Figure 1G*). Cdc5 binding to Spc72 depends on a binding motif in serine residues S231 and S232 in the Spc72 sequence (*Snead et al., 2007*). Consistent with this, we found that GFP-tagged Spc72-AA mutant protein, in which both serine residues were substituted by alanine (S231A, S232A), was randomly distributed during mitosis in exponentially growing cells (*Figure 1G*). Similar to that observed for Spc72-GFP in *cdc5-as1* cells after the addition of CMK-C1, Spc72-AA-GFP displayed faster electrophoretic mobility than the wild-type Spc72-GFP in the absence of the inhibitor (*Figure 1H,I*). However, the lack of Cdc5 activity had hardly any considerable effect on the preferential association of Spc72-GFP to the old SPB (*Figure 1C,F*). Therefore, Cdc5 is a determinant of SPB asymmetric segregation that acts either downstream or in parallel to Spc72 during the establishment of the differential SPB inheritance pattern in budding yeast.

Another key factor acting downstream of Spc72 in generating the SPB distribution scheme is Kar9, which asymmetrically loads onto the old SPB during metaphase (*Juanes et al., 2013*). Subsequently, Kar9 is transported to the microtubule plus ends to bind Myo2, a myosin that travels through actin cables toward the bud, promoting old SPB entrance into the daughter cell and establishing the differential SPB fate (*Korinek et al., 2000*; *Liakopoulos et al., 2003*; *Maekawa et al.,*

2003; Yin et al., 2000). Hence, we analyzed the distribution of a super-folder GFP-tagged Kar9 protein (Kar9-sfGFP) on cdc5-as1 cell SPBs. As with Spc72-GFP, inhibiting Cdc5-as1 also led to randomization of Kar9-sfGFP loading on SPBs in asynchronous cell culture (Figure 2A–C). In contrast, here, the lack of Cdc5 activity perturbed the preferential association of Kar9-sfGFP with the old SPB (Figure 2A,D). Therefore, an important role of Cdc5 in establishing SPB inheritance is to retain specific loading of Kar9 onto the old SPB.

Besides the SPBs, Kar9 displays other subcellular localizations during the cell cycle, being observed both on cytoplasmic and kinetochore microtubules and is also transported between the nucleus and the cytoplasm. It has been suggested that Kar9 asymmetry on cytoplasmic microtubules is partially dependent on protein modifications that take place within the nucleus, where Kar9 is both sumoylated and ubiquitylated (Kammerer et al., 2010; Leisner et al., 2008; Schweiggert et al., 2016). We therefore carried out a broader analysis of Kar9 distribution in the absence of Cdc5 activity. Cdc5-as1 inhibition not only disrupted its preferential polarization toward the old SPB and the daughter cell, but also had a severe impact on the overall localization of Kar9. It further caused increased Kar9-sfGFP nuclear retention and localization to the interpolar microtubules (Figure 2E–G). The change in the Kar9 distribution pattern was also observed when Cdc5 was inactivated by cdc5-2 and cdc5-77 and with mCherry-tagged Kar9 (Figure 2—figure supplement 1A–C).

Lastly, we analyzed whether Kar9 could be post-translationally modified in a Cdc5-dependent manner. Evaluation of the electrophoretic mobility of 13Myc-tagged Kar9 on polyacrylamide gel electrophoresis (PAGE) indicated the accumulation of slow-migrating forms of the protein that were partially dependent on Cdc5 activity (Figure 2H, Figure 2—figure supplement 1D,E). Together, these observations support the premise that Cdc5 can collaborate in establishing SPB fate by promoting post-translational modifications in Kar9 that control its mobilization from the nucleus and its asymmetric distribution to the old SPB.

## Cdc5 regulates efficient Spc72 recruitment on the SPBs

We analyzed the localization of Cdc5 and Spc72 throughout the cell cycle to evaluate whether Polo-like kinase activity could control other aspects of Spc72 function in SPB inheritance. In cells simultaneously expressing Cdc5-as1-eGFP and Spc72-mCherry, the eGFP-tagged Cdc5-as1 protein was not observed on the SPBs when cells were arrested in G1 after addition of the α-factor mating pheromone (Figure 3A,B, Figure 3—figure supplement 1A). Once the cells were released from the G1 block and allowed to synchronously progress in the cell cycle, Cdc5-as1-eGFP loaded on the old SPB before it was duplicated, and subsequently also on the new SPB. Cdc5-as1 remained symmetrically localized on both SPBs until anaphase, finally slowly disappearing as the cells exited mitosis, first from the SPB retained in the mother cell and then from the daughter-inherited SPB (Figure 3A, B, Figure 3—figure supplement 1A). In contrast, Spc72-mCherry was already present on the SPBs in cells blocked in G1 (Figure 3A,C, Figure 3—figure supplement 1A–E). Then, once the cells were freed from G1 arrest, and despite the observation of Spc72-mCherry on both SPBs after their duplication, it mainly accumulated on the dSPB, remaining preferentially associated to this SPB until late anaphase, when the protein became more symmetrically distributed (Figure 3A, C–E, Figure 3—figure supplement 1A–E; Juanes et al., 2013).

To evaluate the role of Cdc5 activity in controlling its own localization and that of Spc72, we analyzed the distribution of these proteins in cdc5-as1 cells synchronously progressing into mitosis in the presence of CMK-C1. As previously shown (Kitada et al., 1993), inactivating Cdc5 only allowed cells to progress up to late anaphase (Figure 3A). CMK-C1 did not affect Cdc5-as1-eGFP localization, indicating that Cdc5 loading on the SPBs is independent of its kinase activity (Figure 3A,B). Cdc5 activity was also not required for maintaining the Spc72 protein already present on SPBs in G1 cells, which had been loaded on these structures during the prior cell cycle (Figure 3A,C, Figure 3—figure supplement 1B–E). However, inhibiting Cdc5 severely perturbed Spc72-mCherry distribution on the SPBs as the cells progressed into a new cell cycle. First, the overall intensity of the Spc72-mCherry fluorescent signal on the SPBs was significantly reduced, suggesting that the loading of new Spc72 on these structures was compromised in the absence of Cdc5 activity (Figure 3D, Figure 3—figure supplement 1B–E). Additionally, Spc72-mCherry did not symmetrically localize when the cells eventually stopped progressing in anaphase, and instead remained preferentially associated to the dSPB (Figure 3C–E). Finally, the signal emitted by the Spc72-AA-GFP mutant was severely reduced on both SPBs and was similar to that of Spc72-GFP on the mother-cell-retained SPB (mSPB)

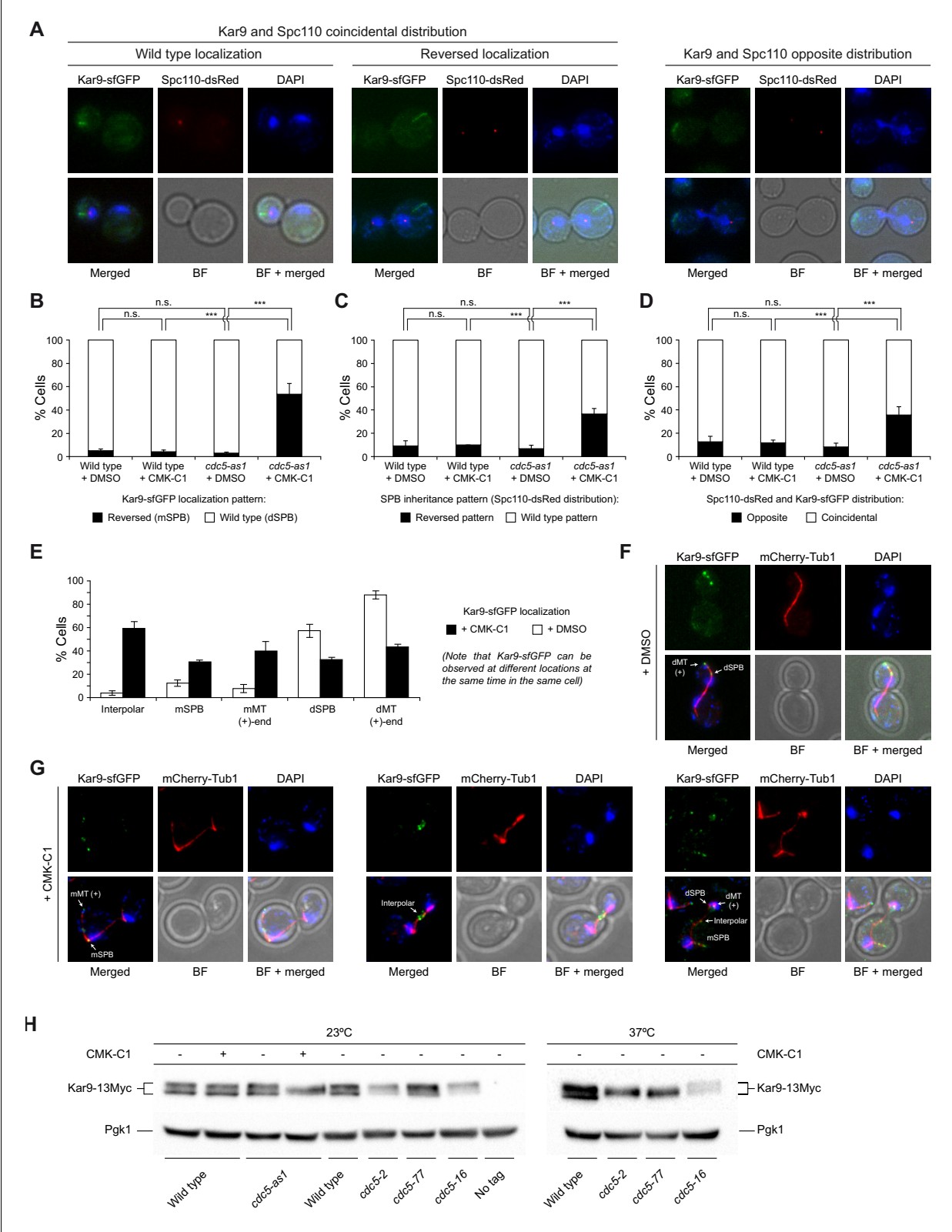

**Figure 2.** Cdc5 regulates Kar9 association with the SPBs. (A–D) Kar9-sfGFP distribution and SPB inheritance in wild-type (F3754) and *cdc5-as1* (F3755) cells also expressing Spc110-dsRed. (A) Representative images showing coincidental or opposite distribution of Kar9-sfGFP (green) and Spc110-dsRed (red), distinguishing in the former between cells exhibiting wild-type and reversed Kar9 localization. Also shown are nuclear morphology (DAPI, blue), BF, and merged images. (B–D) The percentages of cells displaying wild-type (white bars) or reversed (black bars) Kar9-sfGFP localization (B), cells with

*Figure 2 continued on next page*

*Figure 2 continued*

wild-type (white bars) or reversed (black bars) SPB inheritance patterns (based on Spc110-dsRed distribution) (C), and cells showing coincidental (white bars) or opposite (black bars) distribution of both tagged proteins (D). Final data are the average of three experiments (n = 3; 100 cells each). Error bars represent the SD. Statistical significance according to the Newman-Keuls multiple comparison test is indicated. (E–G) Analysis of Kar9-sfGFP distribution in cells expressing mCherry-Tub1 and *cdc5-as1* (F4820). (E) Percentages of cells with Kar9-sfGFP localization to the mSPB or dSPB, to interpolar microtubules, and/or the plus (+) end of mother (mMT) or daughter-attached (dMT) cytoplasmic microtubules, both for cells treated (+CMK-C1, black bars) and untreated (+DMSO, white bars) with CMK-C1. Final data are the average of three experiments (n = 3; 50 cells each). Error bars represent the SEM. (F, G) Representative images of cells illustrating the different localizations of Kar9-sfGFP (green) in the absence (F) or presence (G) of CMK-C1. Also shown are microtubules (mCherry-Tub1, red), nuclear morphology (DAPI, blue), BF, and merged images. (H) Western blot analyses of Kar9-13Myc in asynchronous cultures of wild-type (F3924), *cdc5-as1* (F3964), *cdc5-2* (F4657), *cdc5-77* (F4719), and *cdc5-16* (F4717) cells, and in the wild-type untagged control (F496, No tag). The experiment was performed thrice (n = 3); a representative image is shown. Pgk1 levels were used as the loading control.

The online version of this article includes the following figure supplement(s) for figure 2:

**Figure supplement 1.** Kar9 distribution and post-translational modification in cells lacking Cdc5 activity.

of wild-type cells (*Figure 3F*). Our results indicate that the lack of Cdc5 activity impedes efficient Spc72 loading on the SPBs, interfering with the normal Spc72 distribution pattern and preventing its final symmetric association to the spindle poles (*Figure 4A,B*).

## Cdc5 acts as a molecular timer that facilitates the establishment of SPB fate

We used a bimolecular fluorescence complementation (BiFC) assay to analyze the association of Cdc5 and Spc72 in vivo and how their interaction is modulated throughout the cell cycle (*Sung and Huh, 2007*). To this end, we simultaneously expressed Cdc5-as1 fused to the N-terminal half of the yellow fluorescent protein Venus (Cdc5-as1-VN) and Spc72 linked to the C-terminal half of the same protein (Spc72-VC) in cells synchronously progressing in the cell cycle. The appearance of SPB-associated fluorescent signals resulting from the reconstitution of the Venus molecule demonstrated the in vivo interaction of Cdc5 and Spc72 (*Figure 4C*). However, despite both proteins being present on the SPBs before their duplication (*Figure 3B,C*), the Cdc5–Spc72 association was specifically promoted during metaphase–anaphase transition (*Figure 4D,E*). Furthermore, their interaction was preferentially detected on the dSPB, similar to that observed for the differential distribution of Spc72, although the Cdc5–Spc72 association was asymmetrically favored on this SPB for a longer time (*Figures 3C* and *4E*). Interestingly, although the Cdc5–Spc72 interaction was independent of Polo-like kinase activity, inhibiting Cdc5 compromised their preferential association on the dSPB (*Figure 4F,G*).

Our results suggest that Cdc5 specifically exerts control on Spc72 during metaphase–anaphase transition. To test this hypothesis, we used a conditional mutant of the anaphase-promoting complex cofactor Cdc20 to evaluate how inhibiting Cdc5 activity at different cell cycle stages would affect Spc72 localization (*Muñoz-Barrera et al., 2015*). First, we preserved Cdc5 activity as cells were released from G1 block into subsequent metaphase arrest caused by conditional inactivation of Cdc20, and then inhibited Cdc5 shortly before Cdc20 was reactivated and cells were allowed to enter anaphase. In these cells, the normal Spc72 distribution pattern remained severely affected, as predicted if Cdc5 were required for the efficient recruitment of Spc72 to SPBs as they transitioned from metaphase to anaphase (*Figure 5A,B*, *Figure 5—figure supplement 1A–D*). In contrast, and also according to our hypothesis, the regular pattern of Spc72 localization at this cell cycle stage was eventually established when Cdc5 was initially inhibited as the cells progressed from G1 into metaphase arrest, and then reactivated only shortly before they were allowed to advance again into anaphase (*Figure 5C,D*, *Figure 5—figure supplement 2A–D*). Importantly, the results were not a consequence of comparing cell populations at different cell cycle stages, as blocking cells in anaphase at the end of the experiment with a *cdc5-as1 cdc14-1* background yielded a similar outcome (*Figure 5—figure supplement 3A,B*).

As shown earlier, Spc72 remained asymmetrically loaded on the dSPB after cells were released from G1 arrest and synchronously progressed through the cell cycle in the absence of Cdc5 activity (*Figure 3C–E*). Under these conditions, the lack of Cdc5 activity did not perturb Kar9 bias toward the old SPB (*Figure 2—figure supplement 1F,G*). The normal asymmetric SPB inheritance pattern

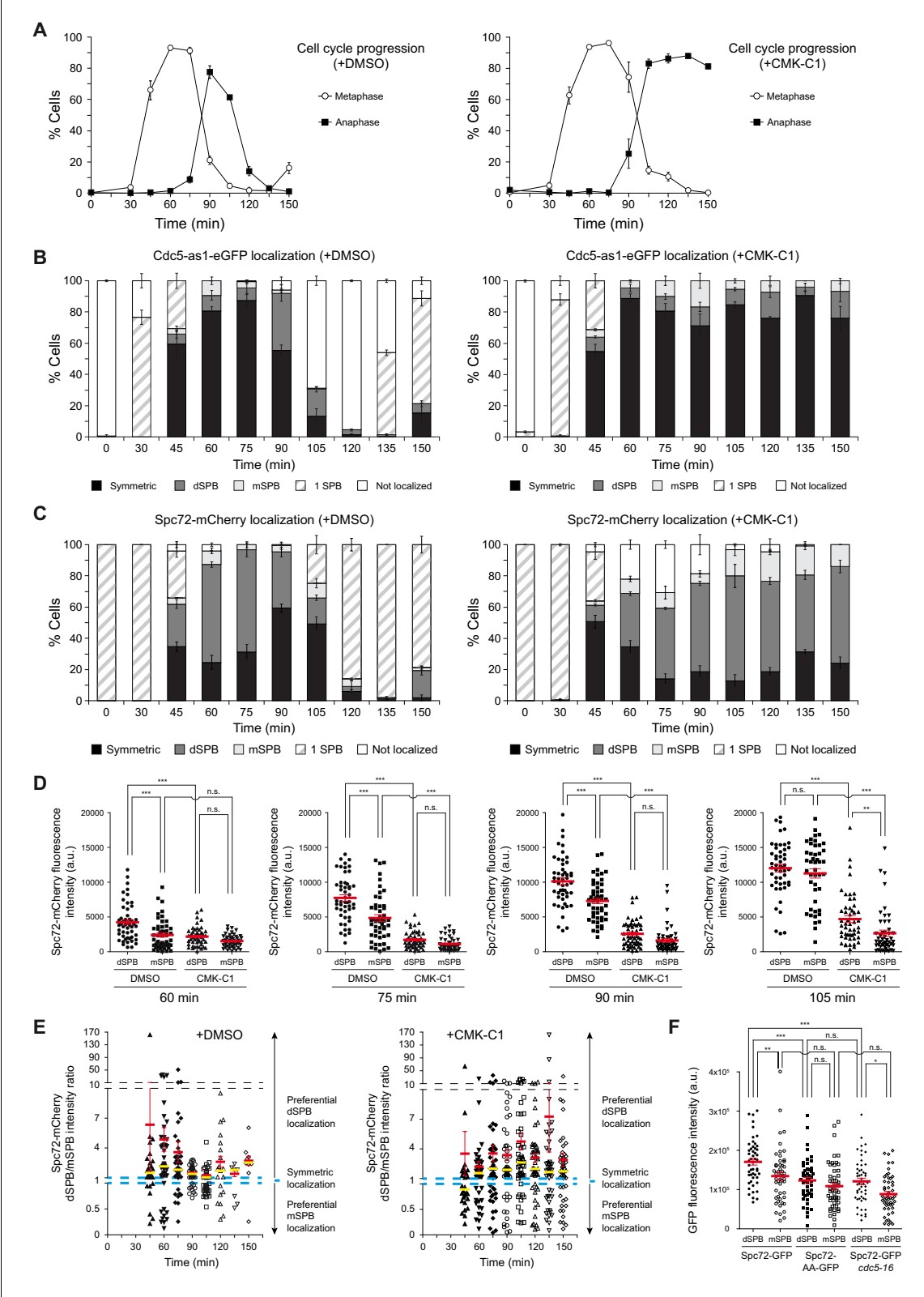

**Figure 3.** Cell cycle analysis of Cdc5 and Spc72 localization. (A–E) Cells expressing Cdc5-as1-eGFP and Spc72-mCherry (F3053) were arrested in G1 with α-factor and released at 26°C in YPAD medium with (+CMK-C1) or without (+DMSO) CMK-C1. The experiment was repeated twice (n = 2), with similar results; a representative experiment is shown. (A) Cell cycle progression according to spindle and nuclear morphologies. The percentages of metaphase and anaphase cells are indicated. Final data are the average of three samples (n = 3; 100 cells per each). Error bars represent the SD. (B, C)

*Figure 3 continued on next page*

*Figure 3 continued*

The percentages of cells displaying localization or non-localization (white bars) of Cdc5-as1-eGFP (**B**) or Spc72-mCherry (**C**) to the SPB before its duplication (one SPB, striped bars), or subsequently to both (symmetric, black bars) or preferentially one SPB. In the latter, it is indicated whether the protein was asymmetrically distributed to the dSPB (dark grey bars) or the mSPB (light grey bars). Final data are the average of three samples (n = 3; 50 cells each). Error bars represent the SEM. (**D**) Scatter plot of Spc72-mCherry fluorescence intensity in the dSPB and mSPB from metaphase to anaphase (60–105 min), in CMK-C1–treated (CMK-C1) or untreated (DMSO) cells. Red bars indicate the mean ± SEM (n = 50 cells). Statistical significance according to the Newman-Keuls multiple comparison test is indicated. (**E**) Scatter plot displaying the dSPB/mSPB Spc72-mCherry fluorescence intensity ratio throughout the time course in CMK-C1–treated (+CMK-C1) or untreated (+DMSO) cells. The ratio equals one when Spc72 is evenly loaded on both SPBs (dashed blue line), >1 when the protein is asymmetrically localized on the dSPB, and <1 when it is preferentially found on the mSPB. The Y-axis is divided into three different scales to facilitate visualization of all samples. Red bars indicate the mean ± SEM and yellow bars the median (n = up to 50 cells displaying two SPBs at each time point). (**F**) Scatter plot of GFP fluorescence intensity in the dSPB and mSPB in asynchronous wild-type (F3696) and *cdc5-16* (F4699) cells expressing Spc72-GFP, and in wild-type cells expressing Spc72-AA-GFP (F4606) during anaphase. Red bars indicate the mean ± SEM (n = 50 cells). Statistical significance according to the Newman-Keuls multiple comparison test is indicated.

The online version of this article includes the following figure supplement(s) for figure 3:

**Figure supplement 1.** Spc72 and SPB distribution during a synchronous cell cycle.

was maintained in that situation (*Figure 3—figure supplement 1F–I*), in agreement with Cdc5 inactivation not disrupting the specific association of Spc72 with the old SPB (*Figure 1F*). These results seem to be at odds with the randomized SPB fate observed after inhibiting Cdc5 in asynchronous exponential cell culture (*Figure 1B*). This apparent contradiction can be explained if Cdc5 confers SPB identity by modifying Spc72 in the prior cell cycle, so that the consequences of inhibiting Polo-like kinase activity throughout one cell cycle would only be evident during the following division. In contrast, Cdc5 inactivation in asynchronous exponential culture provided a temporal window during which cells at different cell cycle stages progressed through mitosis with reduced Polo-like kinase activity until they eventually reached final anaphase arrest after complete Cdc5 inhibition. The randomized SPB fate observed, and therefore the aforementioned discrepancy in the results, could be explained by the transient partial inactivation of Cdc5, together with the fact that Cdc5 regulates further downstream events in determining SPB inheritance besides controlling Spc72 association with the SPBs in metaphase–anaphase transition (e.g., the interaction of Kar9 with the old SPB). To test our hypothesis, and to circumvent the limitation that inhibiting Cdc5 arrests the cell cycle in anaphase, we used the *tab6-1* allele of *CDC14*, which encodes a modified Cdc14 phosphatase that bypasses the anaphase block induced in different mutants from the mitotic exit network signaling cascade (*Shou et al., 2001*). *tab6-1* expression in the *cdc5-as1* background allowed a certain population of cells to evade anaphase arrest after the addition of CMK-C1. Similar to that observed in asynchronous culture, evaluation of the SPB fate in *tab6-1 cdc5-as1* cells that synchronously entered mitosis from G1 block in the absence of Polo-like kinase activity indicated that SPB inheritance was randomized during the following anaphase in the cells that managed to escape the initial Cdc5-dependent anaphase arrest and entered a second cell cycle (*Figure 6A,B*).

To reinforce our conclusions, we also evaluated SPB fate in cells expressing both the *cdc5-as1* and *spc72-AA* alleles. In this background, even without CMK-C1, Cdc5 could not phosphorylate the mutant Spc72 on the SPBs, and the fate of these structures could not be established during the current cell cycle. Therefore, the initial scenario would be similar to that of *tab6-1 cdc5-as1* cells that have gone through a 'first cycle' of Cdc5 inhibition. Hence, after pheromone-induced G1 arrest of the *cdc5-as1 spc72-AA* cells and their subsequent synchronous release in the presence of CMK-C1, we observed a synergistic effect of Cdc5-as1 inactivation with the *spc72-AA* mutation. This resembled the results during a second cell cycle in our prior synchronization experiments with *tab6-1 cdc5-as1* cells. Indeed, while the asymmetric SPB distribution was preserved when Cdc5-as1 was not inhibited, SPB fate was greatly affected as a consequence of the lack of Cdc5 activity in the *spc72-AA* background (*Figure 5—figure supplement 3C*). This evidence forms a strong basis for supporting the role of Cdc5 in conferring the old SPB its identity during the previous cell cycle.

To explain why SPB inheritance was not affected during the first cell cycle after Cdc5 inhibition, it is important to highlight that although this kinase facilitates efficient Spc72 recruitment to both SPBs, it is not required for its maintenance on these structures (*Figure 3A,C*). Consequently, in synchronized cells entering the cell cycle without Cdc5 activity, the old SPB, which already carries Spc72 protein loaded during the previous division, would promote microtubule nucleation more efficiently

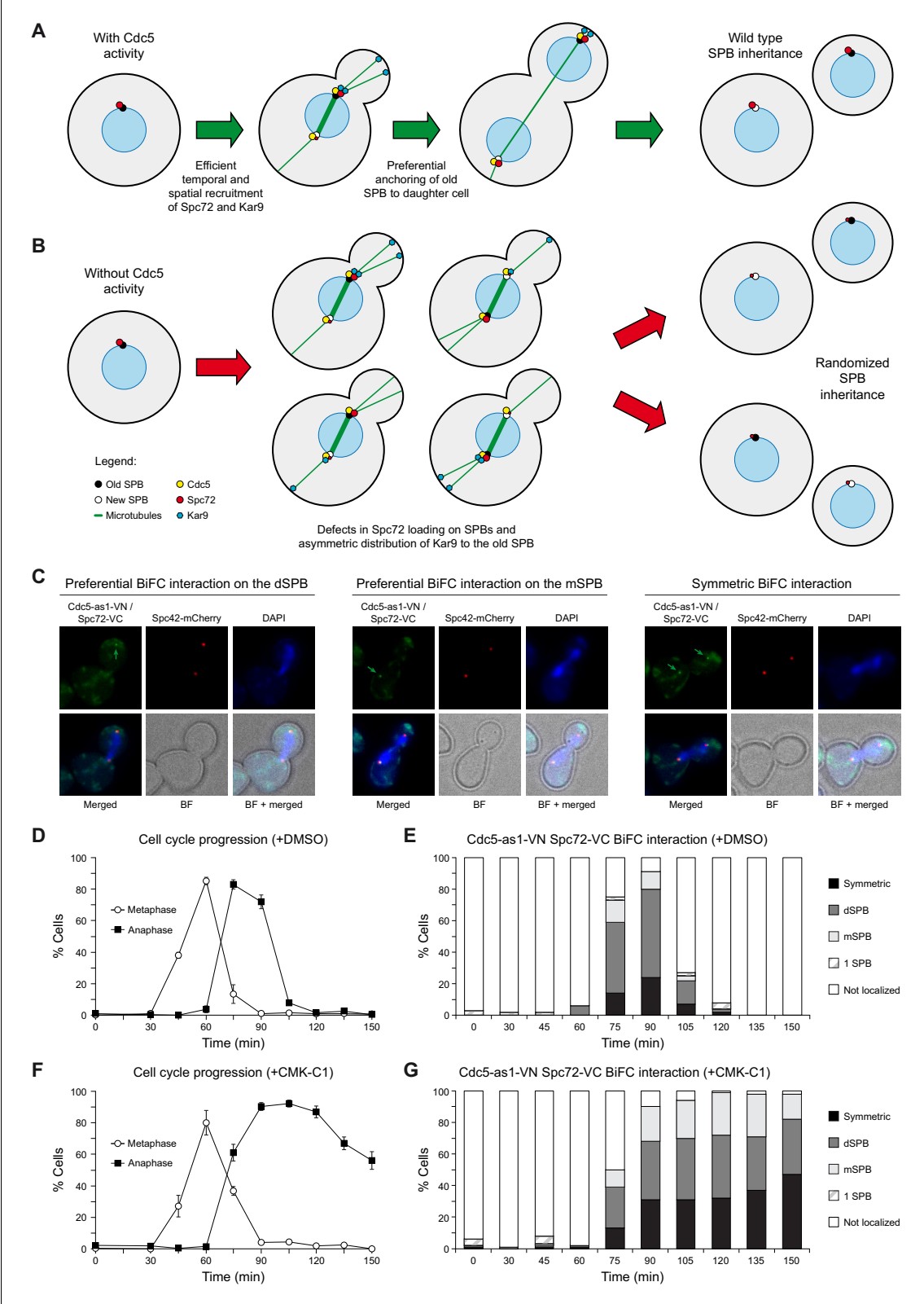

**Figure 4.** BiFC analysis of Cdc5 and Spc72 interaction. (**A**, **B**) Model depicting the role of Cdc5 in the control of Spc72 and Kar9 function on SPB distribution (**A**) and the consequences of the lack of its kinase activity on the predetermined SPB inheritance pattern in budding yeast (**B**). (**C–G**) Cells expressing Cdc5-as1-VN and Spc72-VC (F3420) were arrested in G1 with α-factor and released at 26˚C in YPAD medium with (+CMK-C1) or without (+DMSO) CMK-C1. The experiment was repeated twice (n = 2), with similar results; a representative experiment is shown. (**C**) Representative images

*Figure 4 continued on next page*

*Figure 4 continued*

displaying BiFC interaction and reconstitution of Venus fluorescence (Cdc5-as1-VN/Spc72-VC, green) and Spc42-mCherry fluorescence (red) distribution. The green arrow marks the SPB where the BiFC association is observed. Also shown are nuclear morphology (DAPI, blue), BF, and merged images. (D, F) Cell cycle progression according to spindle and nuclear morphologies. The percentages of metaphase and anaphase cells are indicated. Final data are the average of three samples (n = 3; 100 cells each). Error bars represent the SD. (E, G) Percentages of cells displaying or not displaying (non-localized, white bars) Venus fluorescence on the SPB before its duplication (one SPB, striped bars), or subsequently, on both (symmetric, black bars) or preferentially one SPB. In the latter, it is indicated whether the Venus fluorescent signal was asymmetrically distributed to the dSPB (dark grey bars) or the mSPB (light grey bars) (n = 100).

and also be preferentially inherited by the daughter cell. Hence, Cdc5 would act as a molecular timer that delays the incorporation of new Spc72 onto the SPBs, favoring the interaction of microtubules from the old SPB with the daughter cell cortex and restricting the microtubule nucleation capacity of the new SPB until metaphase onset. This facilitates the establishment of the pre-determined SPB inheritance pattern. Accordingly, the distribution of γ-tubulin (Tub4-mScarlet), which preferentially accumulates on the dSPB up to metaphase–anaphase transition in an unperturbed cell cycle (*Juanes et al., 2013*), was randomized when Cdc5 was inhibited, suggesting similar microtubule nucleation capacity of both SPBs under these conditions (*Figure 5—figure supplement 3D*).

To validate our hypothesis, we released G1-arrested cells into fresh medium containing nocodazole. In the presence of this microtubule-depolymerizing drug, the cells were arrested in metaphase due to spindle assembly checkpoint activation (*Musacchio and Salmon, 2007*). When nocodazole was washed out from the medium, the microtubules could polymerize again and mitosis proceeded once the cells had properly built a bipolar spindle. However, the transient microtubule depolymerization randomized SPB fate (*Figure 6C,D*; *Pereira et al., 2001*). This can be explained by the fact that, in nocodazole-induced metaphase arrest, Cdc5 localizes on both SPBs and can mediate Spc72 loading, promoting microtubule nucleation from these structures after nocodazole wash-out and consequently increasing the chances that microtubules from either SPB are captured by the bud cortex. Accordingly, CMK-C1-untreated *cdc5-as1* cells displayed randomized Spc72 localization on SPBs after release from the nocodazole block (*Figure 6E*). However, inhibiting Cdc5 activity before the cells were allowed to re-establish the mitotic spindle should prevent new Spc72 molecules from being loaded on the SPBs. Consequently, the old SPB would nucleate more microtubules and thereby be preferentially inherited by the daughter cell. In accordance with this prediction, Spc72 remained preferentially loaded on the dSPB, and the normal SPB inheritance pattern was maintained when *cdc5-as1* cells were released from nocodazole arrest in the presence of CMK-C1 (*Figure 6F–H*).

## Cdc5 activity and localization to SPBs are required for controlling Spc72 and Kar9 function

Cdc5 mutants in the POLO-box domain (PBD) cannot localize to SPBs but are still viable in budding yeast, despite the important roles played by Cdc5 on these structures (*Ratsima et al., 2016*). To gain further insight into the regulation of SPB inheritance, we analyzed this process in exponentially growing cells carrying the *cdc5-16* PBD mutant allele (*Ratsima et al., 2016*). *cdc5-16* cells displayed a subtler defect in asymmetric SPB distribution than *cdc5* mutants with impaired kinase activity (*Figure 1—figure supplement 1C*). A possible reason for this is either Cdc5-16 could still somehow weakly associate to SPBs, or Cdc5 activity plays a more important role than its localization in maintaining SPB inheritance. Although both the literature (*Botchkarev et al., 2014*) and our own data indicate that Cdc5-16 retains residual capacity for associating with SPBs, we favor the latter of the two options above. In this way, and supporting the premise that Cdc5 localization to SPBs is not essential for establishing their fate, neither Kar9 distribution (*Figure 2—figure supplement 1B*) nor its post-translational modification pattern (*Figure 2H*) were perturbed in *cdc5-16* cells. However, Spc72 distribution and loading to SPBs were severely affected in those conditions, with both spindle MTOCs exhibiting significantly reduced Spc72 levels (*Figures 1G* and *3F*). Likewise, the *spc72-AA* mutant showed similar phenotypes to *cdc5-16* cells, also displaying decreased Spc72 loading on both SPBs (*Figure 3F*) despite SPB fate and preferential distribution of Kar9 to the old SPB being unaffected by the reduced association of Spc72-AA with SPBs (*Figure 1J*, *Figure 2—figure supplement 1C*). These results indicate that while both Cdc5 activity and loading to SPBs are required for

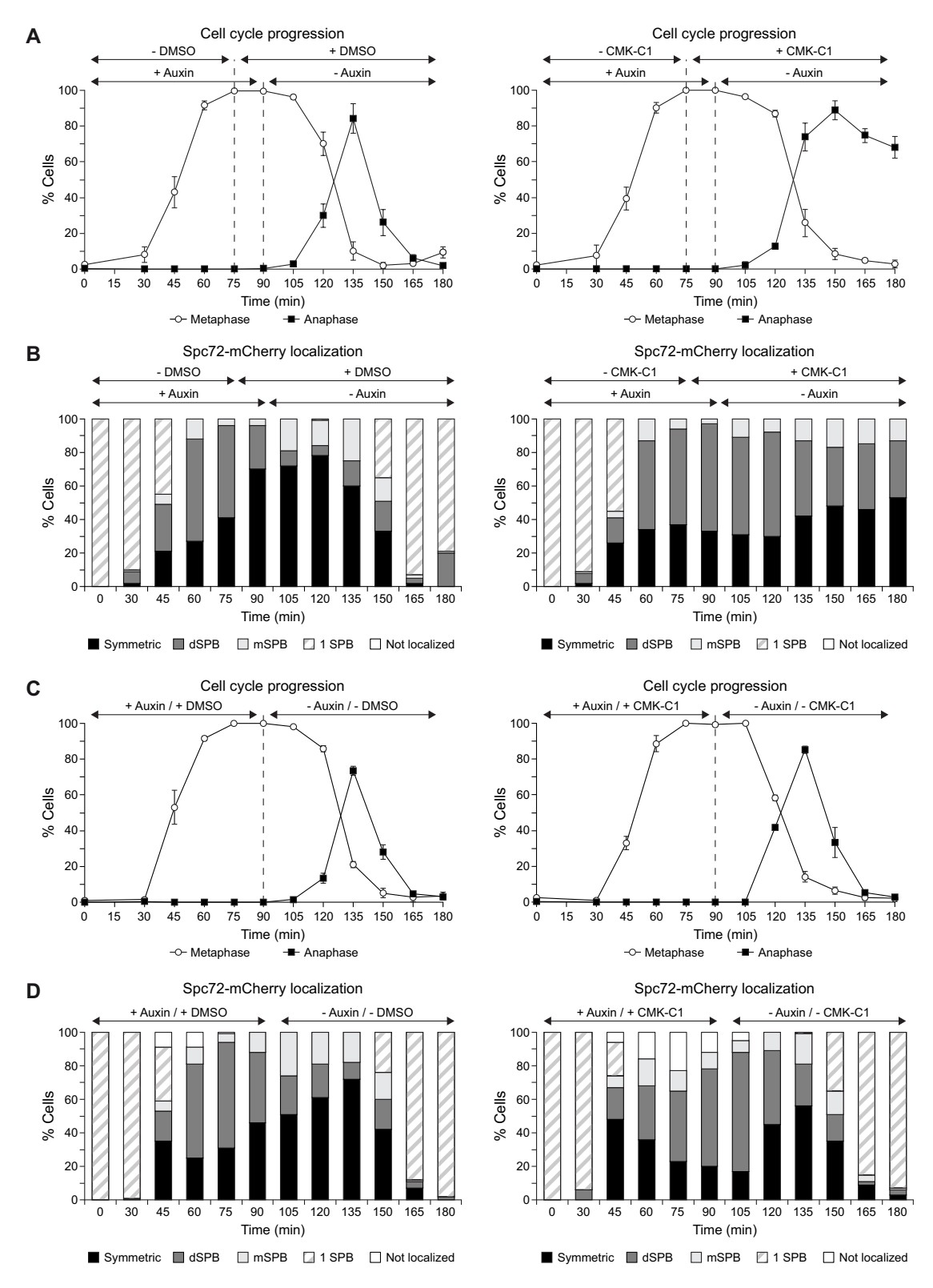

**Figure 5.** Cdc5 activity is specifically required at metaphase–anaphase transition. (A-D) Cells expressing Cdc5-as1-eGFP, Spc72-mCherry, and the conditional *cdc20-AID* allele (F3053) were arrested in G1 with α-factor and released at 26˚C in YPAD with 500 μM indole-3-acetic acid (IAA, +auxin) and with (+CMK-C1) or without (-CMK-C1) CMK-C1. After the metaphase arrest was completed (90 min), the cells were washed out and released in fresh medium without IAA (-auxin) and with (+CMK-C1) or without (-CMK-C1) CMK-C1. To ensure Cdc5-as1 inactivation at metaphase release, 5 μM CMK-C1

*Figure 5 continued on next page*

*Figure 5 continued*

was added 15 min prior to IAA washout (at 75 min) in case it was necessary. As a control, the same experiments were performed with (+DMSO) or without (-DMSO) DMSO instead of CMK-C1. The experiment was repeated thrice (n = 3) in (**A, B**) and twice (n = 2) in (**C, D**), with similar results; a representative experiment is shown in both cases. (**A, C**) Cell cycle progression according to spindle and nuclear morphologies. The percentages of metaphase and anaphase cells are shown. Final data are the average of three samples (n = 3; 100 cells each). Error bars represent the SD. (**B, D**) The percentages of cells displaying or not displaying (white bars) Spc72-mCherry localization to the SPB before its duplication (one SPB, striped bars), or subsequently, to both (symmetric, black bars) or preferentially one SPB. In the latter, it is indicated whether the protein was asymmetrically distributed to the dSPB (dark grey bars) or the mSPB (light grey bars) (n = 100).

The online version of this article includes the following figure supplement(s) for figure 5:

**Figure supplement 1.** SPB inheritance and Cdc5-as1 localization when Cdc5 activity is inhibited after release from a metaphase arrest.

**Figure supplement 2.** SPB inheritance and Cdc5-as1 localization in cells lacking Cdc5 activity only up to the metaphase-to-anaphase transition.

**Figure supplement 3.** Cdc5 acts as a molecular timer that facilitates the establishment of SPB fate.

promoting Spc72 targeting to these structures, only the kinase activity is essential for fostering the pivotal role of Kar9 in maintaining the pre-determined SPB inheritance pattern. These observations also suggest that the loss of Cdc5-dependent regulation of Kar9 has more detrimental consequences on SPB fate determination than that of Spc72.

## Cdc5 activity is required for efficient co-localization of Spc72 and Kar9 to the same SPB

We analyzed the interdependence in Spc72 and Kar9 localization and the potential consequences of the lack of Cdc5 on their efficient co-distribution to the old SPB. According to its pivotal role in establishing the SPB inheritance pattern, and as previously observed (*Moore et al., 2006*; *Pereira et al., 2001*), the lack of Kar9 led to randomized SPB segregation (*Figure 7A*). In contrast, *KAR9* deletion did not interfere with Spc72 loading and preferential association to the old SPB (*Figure 7B,C*), in agreement with Spc72 acting upstream of Kar9 (*Juanes et al., 2013*). *SPC72* deletion, which is viable in the W303 background (*Hoepfner et al., 2002*), also randomized SPB inheritance (*Figure 7D*). However, the lack of Spc72 severely affected Kar9 loading on the SPBs (*Figure 7E,F*), which indicates that a pivotal role of Spc72 in determining SPB fate is mediating Kar9 recruitment to these structures. The lack of Kar9 binding to SPBs in the absence of Spc72 was not due to protein degradation (*Figure 7G*). Furthermore, Kar9 was still post-translationally modified in *spc72Δ* cells (*Figure 7G*), which agrees with our earlier observations, suggesting that Cdc5 likely regulates Kar9 function by promoting its modification in the nucleus and thereby controlling its capacity to be mobilized from this cellular compartment to associate with the old SPB and to polarize toward the daughter cell (*Figure 2H*, *Figure 2—figure supplement 1B,C*).

As stated earlier, and in contrast to Spc72, Cdc5 facilitated the specific association of Kar9 to the old SPB (*Figure 2D*). Additionally, Kar9 localization on the SPBs was more asymmetric than that of Spc72, which could also be observed on the SPB retained by the mother despite being preferentially loaded on the dSPB (*Figures 1C* and *2A*). As Kar9 recruitment to the SPBs depends on Spc72 (*Figure 7E*), specific factors could actively regulate and promote the preferential association of both proteins on the old SPB. To evaluate whether Cdc5 could play a role in favoring Kar9 and Spc72 association, reinforcing the asymmetry in Kar9 distribution, we determined the capacity of Spc72 and Kar9 to interact with each other, and the potential dependence of their association on Cdc5 activity. In accordance with our hypothesis, our analyses demonstrated that both Kar9-13Myc and Tub4-mScarlet co-immunoprecipitated with Spc72-GFP and that the interaction efficiency decreased when the Spc72-AA-GFP mutant was expressed instead of the wild-type Spc72 or when the *cdc5-as1* allele inhibited Cdc5 activity (*Figure 7H,I*, *Figure 7—figure supplement 1A,B*). Finally, we analyzed Kar9 and Spc72 localization in cells expressing *cdc5-as1*. Cdc5 inhibition duplicated the number of cells that showed opposite preferential distribution of Kar9-GFP and Spc72-mCherry (*Figure 7J–L*). Furthermore, the lack of Polo-like kinase activity had a greater impact on the percentage of cells that displayed an inverted Kar9 distribution pattern when Spc72 was correctly localized than on the number of cells in which Spc72 distribution was reversed but Kar9 was localized as in the wild-type. This supports the idea that Cdc5 regulates both Spc72 loading on SPBs and the association of Kar9 with Spc72 and the old SPB (*Figure 7L*). Taken together, these results establish a direct functional link between Cdc5, Spc72, and Kar9.

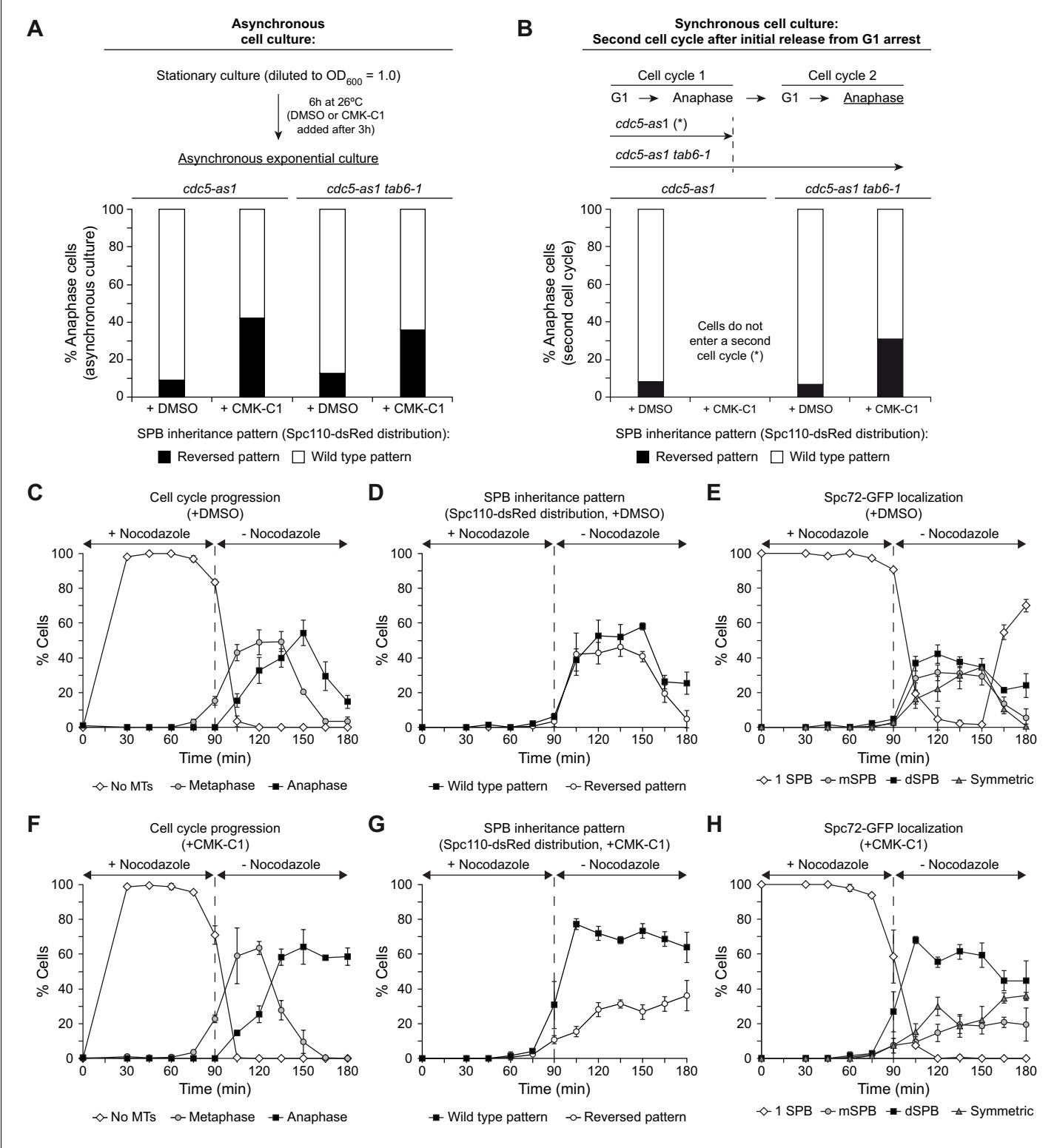

**Figure 6.** Cdc5 activity facilitates recruitment of new Spc72 protein on the SPBs. (A, B) SPB inheritance in *cdc5-as1* (F3705) and *cdc5-as1 tab6-1* (F4140) cells expressing Spc110-dsRed. Graphs show the percentages of cells displaying wild-type (white bars) or reversed (black bars) SPB inheritance patterns, based on Spc110-dsRed distribution, in asynchronous cultures (A) and in cells arrested in G1 with α-factor and released at 26°C in YPAD with (+CMK-C1) or without (+DMSO) CMK-C1 (B). Experiments (A) and (B) were repeated twice (n = 2; 100 cells each) and thrice (n = 3; 100 cells each), respectively, with similar results. A representative experiment is shown in each case. (C–H) Cdc5-as1 cells expressing Spc72-GFP and Spc110-dsRed (F3705) arrested

*Figure 6 continued on next page*

*Figure 6 continued*

in G1 with α-factor and released at 26°C in YPAD with 15 µg/ml nocodazole (+nocodazole) and with (+CMK-C1) or without (+DMSO) CMK-C1. After 90 min, the nocodazole was washed out and the cells were released in fresh medium without nocodazole (-nocodazole) and with (+CMK-C1) or without (+DMSO) CMK-C1. The experiment was repeated twice (n = 2), with similar results. A representative experiment is shown. Error bars represent the SD. (C, F) Cell cycle progression according to spindle and nuclear morphologies. The percentages of cells with depolymerized microtubules (No MTs), and metaphase and anaphase cells are shown. Final data are the average of three samples (n = 3; 100 cells each). (D, G) The percentage of cells displaying wild-type or reversed SPB inheritance patterns, based on Spc110-dsRed distribution. Final data are the average of three samples (n = 3; 50 cells each). (E, H) The percentage of cells displaying Spc72-GFP localization to one (one SPB) or both SPBs. In the latter, it is indicated whether Spc72-GFP was preferentially distributed in the mSPB, dSPB, or symmetrically localized (symmetric) on both SPBs. Final data are the average of three samples (n = 3; 50 cells each).

In agreement with new Spc72 loading being dependent on Cdc5 activity, the preferential bias that Spc72 showed toward the old SPB was maintained after *cdc5-as1 tab6-1* cells synchronized in G1 were allowed to enter a second cell cycle in the presence of CMK-C1, despite SPB fate being randomized as a result of Polo-like kinase inhibition (*Figure 7—figure supplement 1C*). However, while Kar9 preferential loading on the old SPB was not affected when cells synchronously progressed through mitosis in the presence of CMK-C1 after initial G1 arrest (*Figure 2—figure supplement 1F, G*), Kar9 association to the old SPB was impaired in *cdc5-as1 tab6-1* cells that managed to enter a second cell cycle (*Figure 7—figure supplement 1D*). Moreover, in contrast to Spc72, Kar9 remained preferentially associated to the old SPB in the return-to-growth experiment after nocodazole block (*Figure 7—figure supplement 1E,F*). This supports our proposal that Cdc5 is an identifying marker in the old SPB during the prior cell cycle.

Overall, our results support a model in which Cdc5 facilitates timely and sequential recruitment of Spc72 to the SPBs, confers each SPB its identity during mitosis, and promotes the specific association of Kar9 to the old SPB (*Figure 4A,B*). This ensures that the asymmetric segregation of the old SPB is maintained during the subsequent cell cycle.

## Discussion

Polo-like kinases belong to a highly conserved family of proteins that regulate multiple aspects of cell division, from mitotic entry to spindle organization, chromosome segregation, mitotic exit, or cytokinesis (*Archambault and Glover, 2009*). Our work highlights another fundamental function of these kinases by demonstrating that Cdc5, the Plk1 homolog in *S. cerevisiae*, is a key constituent of the cellular machinery that establishes the differential SPB inheritance pattern during the asymmetric division of budding yeast.

Spc72 has a central role in conferring the SPB the capacity to nucleate microtubules (*Knop and Schiebel, 1998*). Before SPB duplication, the old SPB carries Spc72 molecules inherited from the preceding cell cycle. Although Cdc5 phosphorylates Spc72 both in vitro and in vivo, the precise regulatory function of the phosphorylation has not been characterized (*Maekawa et al., 2007*; *Snead et al., 2007*). Here, analyses of the dynamics of Cdc5 and Spc72 localization during the cell cycle demonstrated that the efficient loading of new Spc72 molecules on the SPBs requires both Cdc5 activity and its presence on these structures, and that it is only dynamically promoted once cells reach metaphase–anaphase transition. After SPB duplication, Spc72 preferentially associates with the old SPB up to late anaphase. Likewise, the interaction between Cdc5 and Spc72 is favored in the context of the old SPB. Eventually, however, Spc72 is also loaded to the same extent on the new SPB, becoming symmetrically localized. Interestingly, Cdc5 activity is required for the efficient incorporation of new Spc72 protein on both SPBs and for reaching its final symmetric distribution (*Figure 4A,B*). As Spc72 is loaded on the SPBs in a mainly conservative manner (*Lengefeld et al., 2018*), delaying Spc72 recruitment up to metaphase–anaphase transition determines that microtubule nucleation from the old SPB, and therefore preferential anchoring of the old SPB to the daughter cell cortex (*Juanes et al., 2013*), is initially favored over that from the new one. This contributes to conferring a differential fate to each SPB. On the other hand, the final symmetric localization of Spc72 guarantees that every cell will carry it on the old SPB as they enter the subsequent cell cycle. Finally, and importantly, our results support the premise that Cdc5 activity not only facilitates Spc72

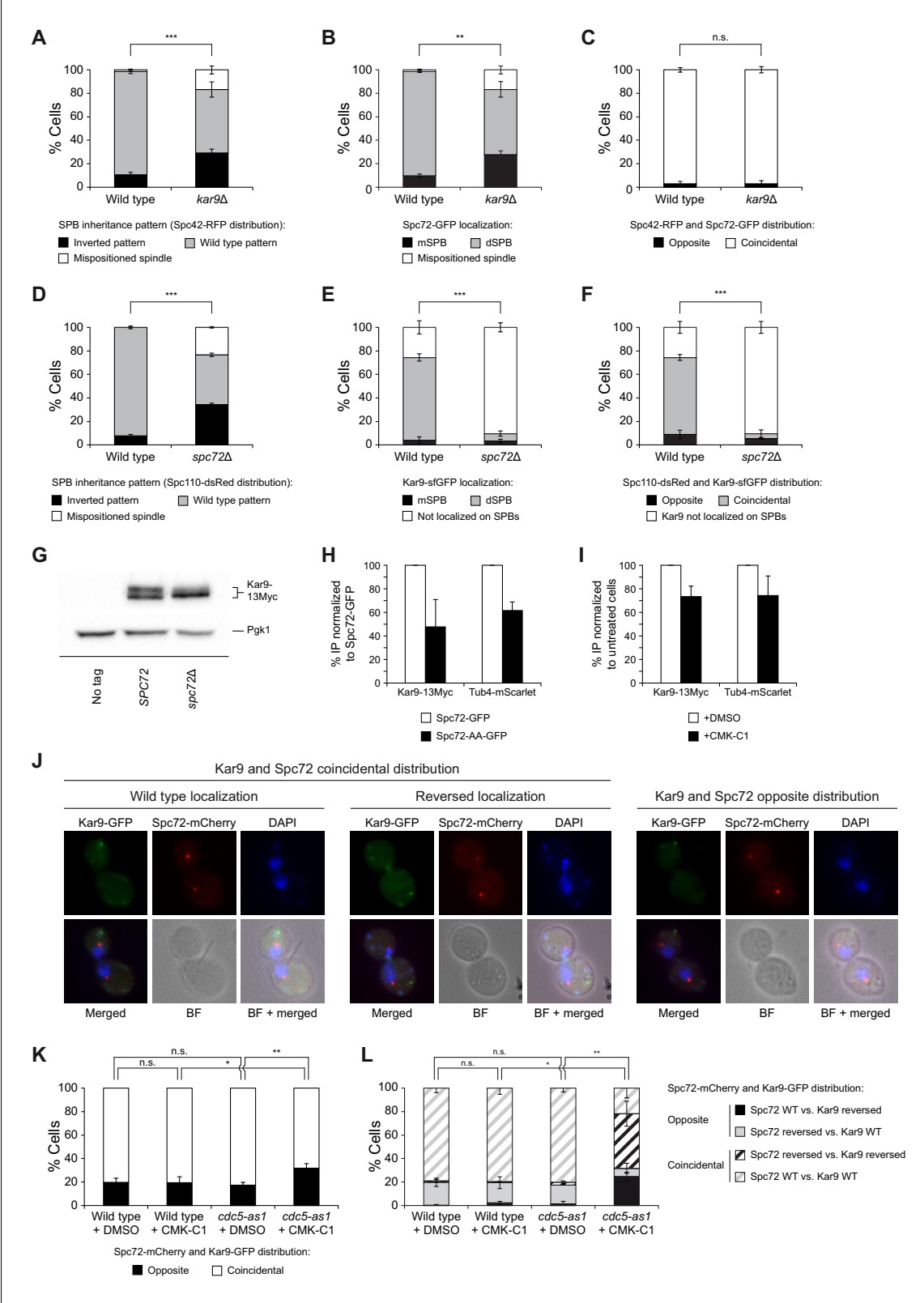

**Figure 7.** Analysis of Spc72 and Kar9 interaction on SPBs. (**A–C**) SPB inheritance (**A**) and Spc72 localization (**B**) in wild-type (F3712) and *kar9*Δ (F3774) cells, both expressing Spc42-RFP and Spc72-GFP. Graphs show the percentages of cells displaying wild-type (grey bars) or reversed (black bars) SPB inheritance patterns, based on Spc42-RFP distribution (**A**), and those of cells exhibiting preferential Spc72-GFP distribution to the dSPB (grey bars) or mSPB (black bars) (**B**). The percentage of cells in which the distribution of these proteins could not be assessed due to spindle misorientation (white

*Figure 7 continued on next page*

*Figure 7 continued*

bars) is further indicated. Also shown are the percentages of cells showing coincidental (white bars) or opposite (black bars) distribution of Spc42 and Spc72 (C). (D–F) SPB inheritance based on Spc110-dsRed distribution (D) and Kar9 localization (E) in wild-type (F3754) and *spc72Δ* (F3836) cells expressing both Spc110-dsRed and Kar9-sfGFP. Graphs show the percentages of cells displaying wild-type (grey bars) or reversed (black bars) SPB inheritance patterns, and those of cells in which the pattern could not be assessed due to spindle misorientation (white bars) (D), and of cells not exhibiting Kar9 on SPBs (white bars) or displaying preferential Kar9-sfGFP distribution to the dSPB (grey bars) or mSPB (black bars) (E). Also shown are the percentages of cells showing coincidental (grey bars) or opposite (black bars) Spc110 and Kar9 distribution and cells with no Kar9 on SPBs (white bars) (F). (G) Western blot analyses of Kar9-13Myc in asynchronous cultures of wild-type (F3924) and *spc72Δ* (F4824) cells, and in the wild-type untagged control (F496, No tag). The experiment was performed thrice (n = 3); a representative image is shown. Pgk1 levels were used as the loading control. (H, I) Percentage of Kar9-13Myc and Tub4-mScarlet protein co-immunoprecipitated with Spc72 in (H) cells expressing either Spc72-GFP (F4825) or Spc72-AA-GFP (F4829) and normalized to those carrying the wild-type GFP-tagged Spc72 (F4825), and in (I) *cdc5-as1* cells expressing Spc72-GFP (F4827) treated with (+CMK-C1) or without (+DMSO) with CMK-C1, and normalized to untreated cells. Final data are the average of three experiments (n = 3). Error bars represent the SEM. (J–L) Kar9-GFP and Spc72-mCherry co-localization in wild-type (F2351) and *cdc5-as1* (F2845) cells. (J) Representative images showing coincidental or opposite distribution of Kar9-GFP (green) and Spc72-mCherry (red), distinguishing in the former between cells exhibiting wild-type (preferential accumulation into the bud) and reversed (retention in the mother cell) distribution patterns of both proteins. Also shown are nuclear morphology (DAPI, blue), BF, and merged images. (K, L) Percentages of cells displaying coincidental (white bars) or opposite (black bars) Kar9-GFP and Spc72-mCherry distribution (K). In each case, it is also detailed whether the proteins showed a wild-type or reversed distribution pattern (L). (A–F, K, L) Final data are the average of three experiments (n = 3; 100 cells each). Error bars represent the SD. Statistical significance according to the Newman-Keuls multiple comparison test is indicated.

The online version of this article includes the following figure supplement(s) for figure 7:

**Figure supplement 1.** Cdc5 establishes SPB identity during the prior cell cycle.

recruitment on the SPBs, but also contributes to bestowing the old SPB its identity during the next cell cycle.

Besides playing a central role in regulating the timely loading and redistribution of Spc72 between the SPBs, Cdc5 also controls the localization and function of Kar9, another key determinant of SPB fate in budding yeast that actively facilitates anchoring of the microtubules emanating from the old SPB in the bud cortex, ensuring segregation of this MTOC toward the daughter cell (*Liakopoulos et al., 2003*). Our results suggest that Cdc5 likely enables asymmetric Kar9 distribution to the old SPB by directly promoting specific cell cycle–dependent post-translational modifications of this protein. An additional and non-mutually exclusive possibility is that Cdc5 can indirectly facilitate the differential association of Kar9 to the old SPB by controlling the activity of factors important for Kar9 loading onto this structure. In contrast to Spc72, only Cdc5 activity, but not its localization on the SPBs, is essential for promoting the pivotal role of Kar9 in maintaining the SPB inheritance pattern. Kar9 is modified by both sumoylation and ubiquitylation (*Kammerer et al., 2010*; *Leisner et al., 2008*). Kar9 is ubiquitylated in the nucleus, where many processes involving SUMO modification also occur, and signaling from kinetochores is important for Kar9 trafficking between the nucleus and cytoplasm and for establishing its asymmetric distribution (*Leisner et al., 2008*; *Schweiggert et al., 2016*). Kar9 sumoylation is controlled by, among other factors, an unknown pathway that is downregulated by the spindle assembly checkpoint (*Leisner et al., 2008*), and Cdc5 is a central target of the main mitotic checkpoints (*Hu and Elledge, 2002*; *Valerio-Santiago et al., 2013*). In agreement with these observations, our data support the premise that the role of Cdc5 in controlling Kar9 function in asymmetric SPB inheritance is exerted in the nucleus and not on the SPBs. Within the nucleus, Cdc5 mediates post-translational modifications of Kar9, regulating its mobilization from the nucleus and its asymmetric distribution to the old SPB and polarization toward the daughter cell. This explains the increased interpolar microtubule localization of Kar9, reminiscent of that in an *xpo1-1* conditional mutant of yeast exportin 1 (*Schweiggert et al., 2016*), and the incorrect Kar9 distribution observed after Cdc5 inactivation.

Our data indicate that Cdc5 controls another aspect of the mechanism regulating SPB inheritance. We demonstrate that Kar9 localization to SPBs is completely dependent on Spc72 and that the association between Kar9 and Spc72 is reduced in the absence of Cdc5 activity. Furthermore, we show that the specific interaction of Cdc5 and Spc72 on the dSPB is compromised by the lack of Cdc5 activity, and that Cdc5 inactivation leads to an increase in the cells that display opposed asymmetric distribution of Spc72 and Kar9 on the SPBs. This evidence suggests that a final contribution of Cdc5 to SPB fate determination is regulating the association of Kar9 and Spc72 to favor Kar9 loading on the old SPB.

Cdc5 also controls Spc72 recruitment to SPBs in the methylotrophic yeast *Ogataea polymorpha*, even though there is uncertainty about whether SPB inheritance is asymmetric in this organism and the differential regulation displayed by Spc72 between both yeasts (*Maekawa et al., 2017*). This analysis suggests that Cdc5 might fulfill other functions in SPB distribution. Our results not only provide insights into the dynamic regulation of Spc72 by Cdc5, but also shed new light on these additional roles of Cdc5 in SPB inheritance. In summary, the evidence collected clearly defines a functional link between Cdc5, Spc72, and Kar9 in SPB fate establishment in *S. cerevisiae* that supports a model in which Cdc5 acts as a molecular timer that promotes the preferential segregation of the old SPB into the daughter cell by controlling the temporal pattern of distribution of both Spc72 and Kar9 during the cell cycle (*Figure 4A,B*). In this way, Cdc5 determines SPB fate by restricting localization of Spc72 to the old SPB, providing a differential microtubule nucleation capacity to each spindle pole. Additionally, Cdc5 promotes Kar9 mobilization from the nucleus, its asymmetric recruitment to the old SPB, and its polarization toward the daughter cell, stabilizing the microtubule array from the old spindle MTOC and ensuring its segregation toward the daughter cell (*Lengefeld et al., 2018*). Furthermore, Cdc5 fine-tunes the asymmetric distribution of the old SPB into the bud by also regulating the association of Spc72 and Kar9. Finally, Cdc5 is required for the proper establishment of SPB identity during mitosis to ensure that the asymmetric segregation of the old SPB can be maintained during the subsequent cell cycle. As Cdc5 regulates Spc72–Kar9 association, a distinct possibility is that it could confer SPB identity by modifying Spc72 so that it facilitates the preferential interaction of Spc72 and Kar9 during the following cell cycle.

Despite Cdc5 being present on both SPBs after their duplication, its association with Spc72 is favored on the old SPB. Hence, other factors are likely to collaborate with Cdc5 in establishing Spc72 asymmetric localization by promoting the interaction of both proteins on the old SPB, facilitating the differential modification and specific cellular destiny of Spc72. One candidate is the Swe1 kinase, which contributes to SPB fate establishment during G1 at least in part through the phosphorylation of Nud1, an integral SPB component that anchors Spc72 (*Lengefeld et al., 2017*). Spc72 and Nud1 interaction is modulated via changes in their phosphorylation status, which both depend on Cdc5 (*Gruneberg et al., 2000*; *Lengefeld et al., 2017*; *Park et al., 2008*). The concerted action of Cdc5 and Swe1 could favor the preferential interaction of Nud1, Spc72, and Cdc5 in the old SPB and the specific association of Spc72 and Kar9 on this particular spindle MTOC, ensuring tight control of SPB inheritance. Accordingly, the Cdc5–Spc72 interaction follows the same temporal pattern as that previously described for Swe1 and Nud1 (*Lengefeld et al., 2017*). Another potential candidate for aiding Cdc5 in conferring a specific cellular fate to each SPB is the cyclin-dependent kinase (CDK) Cdc28. CDK-dependent Kar9 phosphorylation is important for its asymmetric distribution on SPBs (*Liakopoulos et al., 2003*; *Maekawa and Schiebel, 2004*; *Moore and Miller, 2007*). Phosphorylation of certain substrates by Cdc5 requires prior phosphorylation of a POLO-binding domain in the targeted protein by CDK (*Elia et al., 2003*). Hence, CDK-mediated Kar9 phosphorylation could subsequently favor its Cdc5-dependent regulation.

Given the high evolutionary conservation of Polo-like kinases and the proteins that control spindle orientation and spindle MTOC function and distribution during cell division, our results will be helpful for acquiring further knowledge on the molecular mechanisms regulating these processes in higher eukaryotes. Differential spindle MTOC inheritance patterns have been described in neuroblasts and glia progenitor cells from different organisms, establishing a link between non-random centrosome distribution and brain development. During *Drosophila* neuroblast asymmetric division, the centrosome that contains the daughter centriole acts initially as the only functional MTOC and remains stationary, while the centrosome carrying the mother centriole is maintained in an inactive state and does not become proficient as a MTOC until it moves to the opposite side of the cell, which allows the neuroblast to specifically retain the daughter centriole (*Januschke et al., 2011*; *Rusan and Peifer, 2007*). This process depends on the capacity of POLO to phosphorylate the centrosomal protein centrobin (CNB) (*Januschke et al., 2013*). However, neuroblasts expressing a mutant CNB protein that cannot be phosphorylated by POLO retain some specific daughter centriole binding, suggesting that this kinase targets other centrosomal proteins to facilitate their differential behavior (*Januschke et al., 2013*). Interestingly, both POLO and Cnn, the *Drosophila* homologue of Spc72 and human *CDK5RAP2*, localize to the centrosome that retains nucleation activity (*Rusan and Peifer, 2007*). Furthermore, POLO regulates centrosome maturation in *Drosophila* syncytial embryos and human HeLa S3 cells by controlling Cnn and *CDK5RAP* activity,

respectively, although a direct link between this phenomenon and the specification of a differential centrosome fate has not been established (*Conduit et al., 2014*; *Hanafusa et al., 2015*). Based on our results, γ-TuCRs from the Spc72 family are exceptional candidates as mediators of Polo-like kinase function during asymmetric centrosome/SPB distribution. Finally, our data support the analysis of additional undiscovered roles of Polo-like kinases in this process. Correct centrosome orientation in *Drosophila* male GSCs depends on Apc2, the Kar9 and mammalian adenomatous polyposis coli (APC) homolog (*Inaba et al., 2010*; *Yamashita et al., 2003*). Furthermore, APC regulates spindle asymmetry in mouse gut epithelium (*Quyn et al., 2010*), and mammalian APC preferentially localizes to the mother centriole (*Louie et al., 2004*). A key phosphorylation site for Kar9 asymmetric localization to SPBs is conserved in APC (*Trzepacz et al., 1997*). Hence, it would be extremely interesting to analyze whether Plk1 can also control APC localization in human cells, particularly considering that most sporadic colon cancers are attributable to APC mutations (*Thenappan et al., 2009*). Precise understanding of how differential spindle MTOC inheritance is regulated is of utmost importance, especially because the evidence establishes a link between defects in spindle orientation or centrosome distribution during asymmetric cell division with accelerated aging and the development of cancer and neurodegenerative disorders (*Cheng et al., 2008*; *Knoblich, 2010*; *Lancaster and Knoblich, 2012*; *Manzano-López et al., 2019*).

## Materials and methods

### Strains and plasmids

All strains were derivatives of W303 (Table S1, *Supplementary file 1*). Cells were grown in YPD or YPAD (YPD with 300 µg/ml adenine) enriched medium. Strains carrying 13Myc-, mCherry-, mScarlet-, sf-GFP-, and eGFP-tagged fusion proteins were generated by amplifying the corresponding tag sequences using previously described primers (*Longtine et al., 1998*; *Janke et al., 2004*; *Sheff and Thorn, 2004*). Subsequently, the amplification products were integrated by homologous recombination at the C-terminus of the gene, before the stop codon. The *SPC72-AA-GFP* mutant was generated by amplifying the cloned sequence of *SPC72-GFP* using primers containing the S231A, S232A mutation and integrating the amplification product in the wild-type strain by homologous recombination. Strains for BiFC analyses were constructed following the same approach with plasmids described previously (*Sung and Huh, 2007*). Finally, a similar strategy was used for gene deletion, but its endogenous sequence was replaced with a cassette carrying a selectable marker (*Longtine et al., 1998*).

### Cell culture

In experiments where SPB inheritance was estimated by evaluating Spc42-RFP or Spc110-dsRed distribution to these structures, stationary phase cultures in YPAD medium were diluted to optical density at 600 nm ($OD_{600}$) = 1.0 in fresh medium, and grown for 3 hr at 26°C or 23°C (when the *cdc5-2 or cdc5-77* thermosensitive alleles were used). Then, they were treated with (+CMK-C1) or without (+DMSO) 5 µM CMK-C1 at 26°C or 23°C for 3 hr, or grown at 37°C for 3 hr, as indicated in each case.

To analyze Spc72, Kar9, and Tub4 localization and/or post-translational modification, cells were grown overnight in YPAD or YPD medium (for western blot analyses) at 26°C or 23°C (when the *cdc5-2 or cdc5-77* thermosensitive alleles were used). The cultures were then diluted to $OD_{600}$ = 0.2 in fresh medium, grown for 3 hr at 26°C or 23°C, and treated with (+CMK-C1) or without (+DMSO) with 5 µM CMK-C1 for 3 hr at 26°C or 23°C, or grown for 3 hr at 37°C, as indicated in each case.

For synchronous cell cycle analyses, initial stationary or overnight cultures were prepared as indicated earlier according to the type of experiment, but the cells were diluted to $OD_{600}$ = 0.2 in fresh medium and arrested at G1 with 5 µg/ml α-factor.

### Fluorescence microscopy

Fluorescently tagged proteins and DAPI (4′, 6-diamidino-2-phenylindole) staining for nuclear analysis were visualized as previously described (*de los Santos-Velázquez et al., 2017*; *Muñoz-Barrera et al., 2015*) using a DM6000 microscope (Leica) equipped with a 100×/1.40 NA (numerical aperture) oil immersion objective and a DFC350 FX digital charge-coupled device camera (Leica).

Cell images were processed and analyzed with LAS AF (RRID:SCR_013673; Leica) and ImageJ software (RRID:SCR_003070; http://rsbweb.nih.gov/ij/). The fluorescence intensity of Spc72-mCherry was quantified with ImageJ. Z-series images (three planes, ~0.5 μm steps) were captured for each image. Three regions of interest (ROIs) in the same area were drawn and quantified per cell: one around the dSPB, one around the mSPB, and one in the cytoplasm (background signal). For each ROI, we estimated the average intensity of the three planes in the Z-series, subtracting the background intensity from the value obtained for the intensity of the dSPB and mSPB. The fluorescence intensity of Spc72-GFP was quantified using the same approach, but using a maximum intensity projection of the Z-stacks instead of the average intensity.

## Immunofluorescence

Immunofluorescence was performed as detailed previously (*Muñoz-Barrera and Monje-Casas, 2017*) for analyzing cell cycle progression, using specific antibodies at the concentrations described in Table S2 (*Supplementary file 2*). Samples were analyzed and imaged as indicated for visualizing the fluorescently tagged proteins.

## Protein extraction and western blot analysis

Protein extracts were prepared using a trichloroacetic acid (TCA) precipitation method detailed previously (*Cepeda-García, 2017*) or a NaOH extraction method (*Manzano-López et al., 2019*). For TCA precipitation, 10 ml cells from liquid culture were incubated for 10 min in 5% TCA. The samples were centrifuged for 3 min at 1400 rcf and 4°C, and the pellets were transferred to a clean microcentrifuge tube, centrifuged to wash out residual TCA, and resuspended in 1 ml acetone at room temperature using a vortex mixer. The sample was then centrifuged for 7 min at 1400 rcf, and the collected pellets were dried in a hood and resuspended in 125 μl lysis buffer [50 mM Tris-HCl (pH 7.5), 1 mM EDTA, 50 mM DTT, 1 mM PMSF, complete EDTA-free protease inhibitor cocktail (Roche)]. After adding an equal volume of glass beads, the cells were lysed in a vortex mixer for 40 min at 4°C. Finally, 62.5 μl 3 × sample buffer [240 mM Tris-HCl (pH 6.8), 30% Glycerol, 6% sodium dodecyl sulfate (SDS), 600 μg/ml bromophenol blue and 6% β-mercaptoethanol] was added, and the protein extract was boiled for 5 min at 100°C before being loaded in a polyacrylamide gel. For NaOH protein extraction, three $OD_{600}$ units of exponential cell culture were collected and lysed with 400 μl 150 mM NaOH for 5 min at 4°C. The extracts were then centrifuged for 3 min at 3000 × $g$ and 4°C. The supernatant was discarded, and the pellet was resuspended in 150 μl 1 × sample buffer. Finally, the protein extracts were boiled for 5 min before undergoing SDS-PAGE. Western blot analysis of protein levels was performed as detailed previously (*Cepeda-García, 2017*), using specific antibodies at the concentrations indicated in Table S2 (*Supplementary file 2*). The protein expression levels were detected and quantified using WesternBright ECL reagents (Advansta), a ChemiDoc MP system, and Image Lab software (RRID:SCR_014210; Bio-Rad).

## Protein co-immunoprecipitation

Co-immunoprecipitation assays were performed as described previously (*Knop and Schiebel, 1998*), with slight modifications. Briefly, 50 ml exponential yeast culture (~1 × $10^7$ cells/ml) were harvested for each strain, washed once in cold water, and frozen in liquid $N_2$. The pellets were resuspended in 900 μl lysis buffer [20 mM Tris-HCl (pH 7.5), 135 mM NaCl, 2.5 mM KCl] complemented with protease inhibitors [1 mM PMSF, 1 × complete EDTA-free protease inhibitor cocktail (Roche)], and then lysed in a Multi-beads shocker (Yasui Kikai Corporation) for 45 min at 4°C, alternating 60 s pulses at 2500 rpm with 60 s rest. The extracts were cleared twice by centrifugation at 500 × $g$ for 5 min at 4°C to eliminate cell debris. The protein concentration was adjusted by measuring the absorbance at 280 nm with a NanoDrop system (Thermo Scientific). Then, Triton X-100 was added to make up 1% final concentration, and the extracts were incubated at 4°C with rotation for 15 min. After detergent treatment, the extracts were centrifuged twice at 5000 × $g$ for 10 min, and the pellets were discarded. We saved 50 μl extract as the input sample, and 650 μl extract was immunoprecipitated. The input samples were stored at −20°C. For immunoprecipitation, 25 μl agarose GFP-Trap beads (Chromotek) previously equilibrated in lysis buffer with 1% Triton X-100 was added to the samples, and incubated overnight at 4°C with rotation. Subsequently, the samples were washed five times using lysis buffer with 1% Triton X-100. Finally, the immunoprecipitation beads and input samples

were resuspended in 50 µl 1 × sample buffer and warmed at 65°C for 15 min before undergoing SDS-PAGE. Western blot analysis of the protein levels was performed as detailed previously (*Cepeda-García, 2017*), using specific antibodies at the concentrations indicated in Table S2 (*Supplementary file 2*). The protein expression levels were detected and quantified using a Chemi-Doc MP system and WesternBright ECL reagents (Advansta); Kar9-13Myc immunoprecipitation was detected using SuperSignal West Femto substrate (Thermo Scientific).

## Statistical analyses

The figure legends show the details of the statistical analysis for each experiment, including the specific measure used for estimating the variation within each group of data and the exact n value. Statistically significant (***$p<0.001$; **$p<0.01$; *$p<0.05$) or non-significant (n.s.) differences are indicated in the corresponding graphs.

# Acknowledgements

We thank Félix Prado and members of the Monje-Casas' lab for critical reading of the manuscript, José Carlos Blanco-Mira for technical support, Pedro San-Segundo for technical advice, Damien D'Amours and Yasushi Tamura for strains, and Oxford Science Editing for text edition and proof-reading. This work was supported by the European Union (FEDER) and the Spanish Ministry of Economy, Industry and Competitiveness (BFU2013-43718-P and BFU2016-76642-P grants; FPI fellowship to Laura Matellán and Juan de la Cierva research contract to Javier Manzano-López).

# Additional information

## Funding

| Funder | Grant reference number | Author |
|---|---|---|
| MINECO | BFU2013-43718-P | Fernando Monje-Casas |
| European Regional Development Fund | BFU2013-43718-P | Fernando Monje-Casas |
| MINECO | BFU2016-76642-P | Fernando Monje-Casas |
| European Regional Development Fund | BFU2016-76642-P | Fernando Monje-Casas |
| MINECO | FPI fellowship | Laura Matellán |
| MINECO | Juan de la Cierva research contract | Javier Manzano-López |

The funders had no role in study design, data collection and interpretation, or the decision to submit the work for publication.

## Author contributions

Laura Matellán, Javier Manzano-López, Data curation, Formal analysis, Validation, Investigation, Visualization, Methodology; Fernando Monje-Casas, Data curation, Formal analysis, Supervision, Funding acquisition, Validation, Methodology, Writing - original draft, Project administration, Writing - review and editing

## Author ORCIDs

Laura Matellán (iD) https://orcid.org/0000-0002-6897-0148
Javier Manzano-López (iD) https://orcid.org/0000-0002-7149-1690
Fernando Monje-Casas (iD) https://orcid.org/0000-0002-3587-2373

## Decision letter and Author response

Decision letter https://doi.org/10.7554/eLife.61488.sa1
Author response https://doi.org/10.7554/eLife.61488.sa2

## Additional files

### Supplementary files

• Supplementary file 1. Table S1 including all the strains from the *S. cerevisiae* W303 genetic background used in this work.

• Supplementary file 2. Table S2 detailing the list of antibodies used for immunofluorescence and Western blot.

• Supplementary file 3. Excel document including spreadsheets with the source data for each of the figures of the article.

• Transparent reporting form

### Data availability

All data generated or analyzed during this study are included in the manuscript and supporting files. No new datasets were generated or previously published datasets used in our work. Rich media files such as videos, audio clips or animations were also not created for this article. No further relevant additional data file was provided.

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
