## [Decision Letter]

**Acceptance summary:**

Your study provides important insights into how microtubule organizing centers, known as centrosomes or spindle pole bodies (SPBs), are asymmetrically inherited during the cell cycle in budding yeast. In demonstrating the involvement of an essential cell cycle kinase, Cdc5, in this process and by delineating two of its downstream effectors, Spc72 and Kar9, your study elucidates key regulatory steps that controls the asymmetric inheritance of centrosomes/SPBs. These findings have broad implications for processes such as development and aging.

**Decision letter after peer review:**

Please take note of the points below. Specifically, we would like you to address 1B, 2, 3 and 4 before returning your submission. We hope you will continue to support *eLife*.

The study by Matellán et al. investigates how asymmetric inheritance of the SPB in budding yeast, the equivalent of the centrosome in animal cells, is regulated. They identify the Polo-like kinase Cdc5 as a crucial regulator of this process, by promoting the sequential recruitment of two asymmetry determinants to the SPBs, the γ-tubulin complex adapter Spc72 and the downstream effector Kar9. They propose a model whereby the old SPB, carrying Spc72 from the previous cell cycle, has a larger time window starting early in mitosis to preferentially segregate into the daughter cell. Only later during mitosis Cdc5 promotes recruitment of additional Spc72, generating a symmetric distribution between SPBs. This ensures that after cell division each cells receives an SPB pre-loaded with Spc72.

Overall this is a nice paper from an expert laboratory with previous contributions to the field. The mechanisms underlying asymmetric inheritance of SPBs/centrosomes have important implications for processes such as development and aging and remain a controversial subject of study. The authors make an important contribution to this topic. The experiments are well designed and support the conclusions. In general, I support publication in *eLife*, but I have a few comments that should be addressed:

1) I found the manuscript quite hard to read and follow. It seems that this issue was brought up before, but I think there still is room for improvement.

a) In the results part, the authors may want to consider breaking up some very long paragraphs into multiple, shorter, more concise paragraphs. Ideally each paragraph would end with one summarizing sentence that states the main conclusion.

b) Due to the complexity of the data and its presentation, by the end of the Results section non-expert readers may have lost track of what the main message of the paper is. In this regard, the first two paragraphs of the Discussion are a nice summary of the findings. I suggest adding a visual representation in the form of a cartoon model.

c) In my opinion the Discussion is too long. Parts that do not directly deal with the presented data should be kept much shorter.

2) The quality of the western blot in Figure 1H is not sufficient to show the claimed mobility shifts. The added line profiles are difficult to read. It is not indicated where they were drawn and what the y-axis represents. Also there is only one lane "+ CMK-C1" in the western, but two curves in the "+ CMK-C1" plot.

3) One cannot see mobility shifts in Figure 1—figure supplement 1E. Instead there seem to be variable intensities of the Spc72-13Myc band, but these seem similar for both conditions.

4) Figure 3E: I strongly suggest indicating by red bars the median and not mean values. There are extreme outliers and the plotted means poorly represent the bulk of the data points.

---

## [Author Response]

[…] In general, I support publication in eLife, but I have a few comments that should be addressed:1) I found the manuscript quite hard to read and follow. It seems that this issue was brought up before, but I think there still is room for improvement.a) In the results part, the authors may want to consider breaking up some very long paragraphs into multiple, shorter, more concise paragraphs. Ideally each paragraph would end with one summarizing sentence that states the main conclusion.b) Due to the complexity of the data and its presentation, by the end of the Results section non-expert readers may have lost track of what the main message of the paper is. In this regard, the first two paragraphs of the Discussion are a nice summary of the findings. I suggest adding a visual representation in the form of a cartoon model.c) In my opinion the Discussion is too long. Parts that do not directly deal with the presented data should be kept much shorter.

We thank the reviewer for his/her positive comments about our results and conclusions, as well as for the suggestions to improve the final published version of our article. Following the instructions given during the revision process, the manuscript was extensively edited and underwent language quality control check by a professional language editing service. Nonetheless, we are grateful for the last reviewer’s indications to further facilitate the understanding of the manuscript. In this sense, and in agreement with the editor guidelines, we find particularly helpful the suggestion of introducing a cartoon model to facilitate the interpretation of our conclusions to non-expert readers. This model has been introduced in Figure 4A, B.

2) The quality of the western blot in Figure 1H is not sufficient to show the claimed mobility shifts. The added line profiles are difficult to read. It is not indicated where they were drawn and what the y-axis represents. Also there is only one lane "+ CMK-C1" in the western, but two curves in the "+ CMK-C1" plot.

The mobility shift associated to modifications in Spc72 phosphorylation is indeed difficult to evaluate in Western blots. Spc72 has a considerable size (∼72 Kd, plus the tag size) and it is not heavily modified, which makes changes in the electrophoretic mobility of the protein associated to the lack of Cdc5-dependent phosphorylation hard to detect. This problem has been in fact previously described in the literature, and it is precisely the reason why we added the lane profiles. In order to discern slight changes in Spc72 gel mobility and facilitate evaluation of the results, Maekawa et al., 2017 made use of a graphical representation of the intensities of the bands in the Western gel. We followed the same approach to fairly conclude that *cdc5-as1* cells showed reduced levels of post-translational modification of the Spc72 protein, in agreement with a lack of Cdc5-dependent phosphorylation, and that the same was true for the Spc72-AA mutant (Figure 1H). We do apologize, however, because as the reviewer correctly pointed out there were mistakes in the indicated figures. As such, the Western blot in Figure 1H had a lane corresponding to the wild type Spc72-GFP protein that was not correctly labelled as “+ CMK-C1”. On the other hand, it was not indicated what did the axes represent in Figure 1J. Both mistakes have been corrected.

3) One cannot see mobility shifts in Figure 1—figure supplement 1E. Instead there seem to be variable intensities of the Spc72-13Myc band, but these seem similar for both conditions.

As indicated in the prior comment, mobility shifts due to changes in Spc72 phosphorylation are difficult to evaluate by Western blot. These problems in the detection of electrophoretic mobility are exacerbated in experiments such as the one depicted in the indicated figure, in which the changes are analyzed in a population of cells progressing trough the cell cycle. Although the reviewer is right in that there are variable intensities of the Spc72-13Myc band, it is also evident that the lanes at the indicated time points are wider for the untreated cells than for cells treated with the CMK-C1 inhibitor, suggesting that Spc72 migrates faster in the latter conditions and thus that it is less modified. In any case, and in order for the reviewer to have a more objective assessment of this observation, we include herewith a graphical representation of the quantification of the width of the bands (in pixels) measured from Figure 1—figure supplement 1E (lanes where the mobility shift is observed in the presence of Cdc5 activity are indicated in black).

**Author response image 1. sa1fig1:** 

These results are in agreement with Spc72 being phosphorylated by Cdc5.

4) Figure 3E: I strongly suggest indicating by red bars the median and not mean values. There are extreme outliers and the plotted means poorly represent the bulk of the data points.

As requested by the reviewer, we have indicated the median values in the graphs in Figure 3E. However, we have instead used yellow bars to designate the median and we have still maintained the mean values as red bars to keep the consistency with the rest of the figures of the manuscript, all of which use this statistical parameter as a representation of the average value in each sample.

[Editors' note: we include below the reviews that the authors received from another journal, along with the authors’ responses.]

We are extremely grateful to the reviewers for their positive comments about our work, as well as for their essential contribution to significantly improve the quality of our initial manuscript. Detailed herewith you can find a point by point answer to the concerns raised during the revision process.

Reviewer #1:I think the study by Matellán and colleagues makes a mechanistically important contribution to our understanding of cellular processes responsible for asymmetric centrosome/SPB inheritance during cell division. I am therefore in support of the publication of this study, after the authors implement the revisions listed below.

We thank the reviewer for considering our work. During this revision period, we have made a considerable effort to carry out all the suggested experiments. Please, find below our response to the points raised, which we believed we have managed to answer in a satisfactory manner.

Major comments:The over-reliance on the cdc5-as1 allele to study the role of POLO kinase/Cdc5p in this study is problematic. […]. As a consequence, it is crucial that the authors confirm several of the key results in their study using other, more standard alleles of CDC5.

To provide further support to the results obtained with the *cdc5-as1* allele, we have extended our conclusions by including in our analyses different additional alleles of *CDC5*. Specifically, we have evaluated SPB inheritance, as well as the distribution and post-translational modification profiles of both Spc72 and Kar9, in cells carrying the *cdc5-2* and *cdc5-77* alleles, which encode Cdc5 mutant proteins with reduced kinase activity (Hu et al., 2001, Ratsima et al., PNAS 2011). Remarkably, exponentially-growing cells from both the *cdc5-2* and *cdc5-77* mutants also displayed a randomized pattern of SPB inheritance when shifted at the restrictive temperature (new Supplementary figure 1C), as demonstrated in our initial manuscript for *cdc5-as1* cells after addition of the CMK-C1 inhibitor. These results reinforce our conclusion that Cdc5 activity is necessary for the maintenance of the non-random pattern of SPB distribution in *S. cerevisiae*. Moreover, and further in agreement with our original results, Spc72 distribution was also randomized in *cdc5-2* and *cdc5-77* cells growing in the abovementioned conditions (new Figure 1G). Besides Spc72 localization, we additionally evaluated the changes in the phosphorylation status of Spc72 in *cdc5-2* and *cdc5-77* cells after inhibition of Cdc5 activity. Spc72 has a considerable size (∼72 Kd, plus the tag size) and it is not heavily modified, which makes it difficult to detect changes in the electrophoretic mobility of the protein associated to the lack of Cdc5dependent phosphorylation. In fact, previous analyses of the defects in Spc72 phosphorylation in the absence of Cdc5 activity made use of a graphical representation of the intensities of the bands in the Western gel to discern slight changes in protein gel mobility (Maekawa et al., 2017). Nonetheless, by using the same approach we can fairly conclude that *cdc5-2* and *cdc5-77* cells also showed reduced levels of post-translational modification of the Spc72 protein, in agreement with a lack of Cdc5-dependent phosphorylation (new Figure 1H, I).

Finally, we have confirmed the results regarding the distribution and post-translational modification of Kar9. In our initial manuscript, we exclusively focused on the localization of Kar9 to the SPBs, which we showed to be randomized in exponentially growing *cdc5-as1* cells after CMK-C1 addition (Figure 2B). However, a suggestion made by reviewer #2 led us to the realization that we overlooked a very interesting phenotype regarding Kar9 distribution within the cell, which is also of particular relevance in the context of the next major concern raised by reviewer #1. Specifically, we further noticed that not only Kar9 preferential loading on the old SPB and its asymmetric polarization towards the daughter cell was perturbed as a consequence of the lack of Cdc5-as1 activity, but also that addition to the CMK-C1 inhibitor led to increased nuclear retention and interpolar microtubule localization of Kar9 (new Figures 2E-G). Again, and importantly, results were very similar when Kar9 distribution was analyzed in exponentially growing *cdc5-2* and *cdc5-77* cells when transferred at the restrictive temperature (new Figures 2F, G). Furthermore, evaluation of the electrophoretic mobility of a Kar9-13Myc tagged protein by Western blot analysis clearly demonstrated the disappearance of a band displaying reduced electrophoretic mobility (and therefore more heavily post-translationally modified) after Cdc5 was inhibited in *cdc5-as1*, *cdc5-2* and *cdc5-77* cells (new Figure 2H). In summary, all the previously detailed experiments using different alleles of *CDC5* to inactivate POLO-like kinase activity, strongly reinforce our initial conclusion that Cdc5 collaborates in the establishment of SPB fate in budding yeast by regulating both Spc72 and Kar9 function.

Related to the point above, I was surprised to see the authors did not confirm their central model with polo-box domain (PBD) mutants of CDC5. […]. The Lee, D’Amours and Weinreich groups have clearly shown that Cdc5 alleles carrying inactivating mutations in the PBD are not enriched on SPBs and are viable in yeast. The Shokat group has also shown the interaction between Cdc5p and Spc72p is PBD-dependent in vitro. If the authors' main conclusions are correct, one would predict that PBD-mutants of CDC5 would have SPB fate phenotypes similar to that of the cdc5-as1 mutant […]. Confirming this prediction would validate the main working model of the authors and significantly strengthen their manuscript.

We agree with the reviewer in that a polo-box domain (PBD) mutant of Cdc5 that cannot localize to the SPBs does indeed provide an excellent tool to refine our model. Hence, we have completed the analyses described in the previous major point by expanding our studies using cells carrying the *cdc5-16* PBD mutant allele (Ratsima et al. PNAS 2011). Remarkably, when compared with the *cdc5* mutants impaired in POLO kinase activity, exponentially-growing *cdc5-16* cells displayed a subtler phenotype in terms of their defects in asymmetric SPB inheritance (new Supplementary figure 1C). This could be explained either if Cdc5-16, despite being severely compromised in SPB loading, could still weakly associate to these structures and maintain SPB fate or, alternatively, if Cdc5 kinase activity but not its localization to the SPBs was necessary to establish the predetermined SPB inheritance pattern. Although both previously published (Botcharev et al., 2014) and our own data indicate that Cdc5-16 still retains some residual capacity to associate with the SPBs, additional results obtained during the revision of our manuscript made us to instead favor the last of the two possibilities. In this way, and supporting that SPB localization of Cdc5 is not essential for establishing the fate of these MTOCs, neither Kar9 distribution (new Figure 2F) nor its post-translational modification pattern (new Figure 2H) were perturbed in *cdc516* cells growing at the restrictive temperature. Remarkably, however, Spc72 loading to the SPBs was severely affected in these cells, with both MTOCs displaying significantly reduced levels of the γ-TURC (new Figures 1G, 3F). These exciting new results indicate that while both Cdc5 activity and loading to the SPBs are required to promote Spc72 targeting on the SPBs, only the activity of the POLO-like kinase is essential to foster Kar9 pivotal role in the maintenance of the pre-determined SPB inheritance pattern. Moreover, these observations also suggest that the loss of Cdc5-dependent regulation of Kar9 has more detrimental consequences on the determination of SPB fate than that of Spc72.

Interestingly, it has been shown that Kar9 is modified both by SUMOylation and by ubiquitination (Leisner et al., 2008; Kammerer et al., 2010). The ubiquitination of Kar9 takes place in the nucleus, where many of the processes involving SUMO modification also occur, and signaling from kinetochores was proposed to be important for Kar9 trafficking between nucleus and cytoplasm, as well as for the establishment of its asymmetric distribution (Leisner et al., 2008; Schweiggert et al., 2016). In fact, SUMOylation of Kar9 was shown to be controlled, among other factors, by an unknown pathway that is downregulated by the spindle assembly checkpoint (Leisner et al., 2008). Since Cdc5 is a central target of the main mitotic checkpoints (Hu et al., 2001; Valerio-Santiago et al. PLoS Genetics 2013), it is therefore feasible that the specific role of the POLO-like kinase in regulating Kar9 function in asymmetric SPB inheritance was exerted in the nucleus, and not on the SPBs. In this way, Cdc5 could be mediating post-translational modifications of Kar9 in the nucleus, thus interfering with mobilization of the protein from this cellular compartment and also with its polarization towards the daughter cell. This would explain the increased interpolar microtubule localization of Kar9, reminiscent of that in a *xpo1-1* conditional mutant of yeast exportin 1 (Schweiggert et al., 2016), and the incorrect asymmetric distribution of Kar9 that we observed after Cdc5 inactivation (new Figures 2E-G).

To further strengthen the previous exciting new results, which give additional support to our proposal that Cdc5 plays many roles in the establishment of the SPB inheritance pattern, we generated a Spc72-AA mutant protein lacking two serine residues in the γ-TURC that are bound by Cdc5 (S231A, S232A) (Snead et al., 2007). Remarkably, as expected according to our hypothesis, Spc72-AA-GFP was randomly distributed during cell division, as observed for the wild type protein after impairing Cdc5 kinase activity or its localization to the SPBs (new Figure 1G). Furthermore, the *spc72-AA* mutant showed similar phenotypes to those observed for *cdc516* cells. As such, Spc72-AA displayed reduced levels of loading on both SPBs (new Figure 3F). However, and as also anticipated, SPB fate and preferential distribution of Kar9 to the old SPB were not affected by the reduced association of the Spc72-AA mutant with the SPBs (new Figures 1J and 2G). In summary, our results support a model according to which Cdc5 activity and localization to the SPBs is necessary to facilitate the efficient timely and sequential recruitment of Spc72 to these structures and thus to favor the segregation of the preexistent SPB into the daughter cells, while only Cdc5 activity is required to enable the specific association of Kar9 to the old SPB, thus decisively contributing to the final establishment of the SPB inheritance pattern.

Several key references directly relevant to this study are missing in the manuscript. For instance, the fact that Cdc5p regulates the recruitment of Spc72p to SPBs has been recently reported by Maekawa et al., 2017. Also, the fact that Cdc5p regulates SPB biogenesis and integrity in budding yeast has been initially observed by Song et al., 2000, Ratsima et al., 2011 and Elserafy et al., 2014.

We did indeed include the reference about the study showing that Spc72 is regulated by Cdc5 in the methylotrophic yeast *Ogataea polymorpha* (Maekawa et al., 2017) in our original manuscript. This analysis, despite the differences observed in the mechanisms controlling nuclear position and spindle orientation between this organism and budding yeast, the uncertainty about whether SPB inheritance is asymmetric in *O. polymorpha* and the authors’ suggestion that Spc72 regulation probably differ between both species (and even unlikely to be controlled by Cdc5 in *S. cerevisiae*), is in fact additionally useful to emphasize important novel aspects of our research. As such, Maekawa et al. already suggested in their article that Cdc5 could fulfill other functions in the regulation of SPB distribution. Our results, besides providing new insights into the dynamic regulation of Spc72 by Cdc5, allow us to further demonstrate that this kinase indeed plays additional roles in SPB inheritance by regulating Kar9 function and bestowing each SPB its cellular fate. We have therefore highlighted this point in the Discussion. As for the rest of the references, although indeed relevant, we apologize because we could not include those and many other important articles due to space constrains. However, and following the reviewer’s indications, we have introduced the suggested bibliography reporting other functions of Cdc5 in SPB biogenesis and integrity in budding yeast.

Minor comments/suggestions:Figure 5 is very difficult to interpret. In particular, it is unclear why the results for the 90 min time points in the graphs in panel B are so different when in fact the two conditions are identical (ie, + auxin / without DMSO or CMK-C1). Can the authors provide an explanation for these inconsistent results? Another reason why this experiment is difficult to interpret is that readers are expected to compare populations of cells that are at dramatically different stages of the cell cycle (G1 vs anaphase) in the 105-180 min timeframe. This experiment should really be conducted in cdc5as1 dbf2-ts or cdc5-as1 cdc15-ts backgrounds because it would synchronize the population without the CMK-C1 inhibitor in anaphase, and thus allow meaningful comparison of Spc72p localization on SPBs with cdc5-as1 strains treated with the inhibitor.

The difference in the 90 min time point from panel 5B is likely due to the fact that we indeed pre -added the CMK-C1 inhibitor 15 min before the metaphase release (*i.e.*, at 75 min). We do apologize because we should have specified this both in the graph and in the legend of Figure 5, which we have now accordingly corrected. We originally did this experiment exclusively in a *cdc5as1* background because of the role that mutants in the Mitotic Exit Network (MEN), such as *dbf2* or *cdc15*, have on SPB inheritance (Hotz et al., Cell 2012) or the function that Cdc14 plays on regulating Cdc5 activity (Lu et al., Cell 2010). In any case, to answer to the reviewer’s concern, we have repeated the experiment again in a *cdc5-as1 cdc14-1* mutant background to compare final populations of cells blocked at the same cell cycle stage (anaphase). As observed in Supplementary figures 5A and B, similar results were obtained in this mutant background, which supports our observations regarding the effect of Cdc5 inhibition on Spc72 loading on the SPBs.

In the experiment described in Figure 6CD: Are the author seeing more loading of Spc72p on the mSPB in the nocodazole arrest? They propose this would be the case, but do not show this is the case. This is an easy experiment that would strengthen their study.

As the reviewer correctly points out, and according to our hypothesis that Cdc5 enables Spc72 loading on both SPBs at the metaphase-to-anaphase transition, once that cells are allowed to exit the metaphase block and repolymerize the mitotic spindle it would be expected to observe more loading of the γ-TuCR on the mSPB. This was indeed what we observed, although the data was not originally included because we thought it was redundant. As now shown in new Figure 6E, cells not treated with the Cdc5-as1 inhibitor showed increased Spc72 protein localized on the mother-retained SPB after release from the nocodazole arrest. However, and in agreement with Cdc5 having a pivotal role in loading of new Spc72 molecules on the SPBs, the γ-TuCR was still preferentially localized to the daughter-inherited SPB in cells treated with CMK-C1 (new Figure 6H). Please, note that addition of nocodazole causes both SPBs to collapse (Monje-Casas and Amon. Dev Cell 2009), which prevented us from comparing the localization of Spc72 on the dSPB and the mSPB until cells repolymerized the mitotic spindle.

In figure S1D, the alleged phosphorylation-induced gel shift of Spc72p in cdc5-as1 cells treated with DMSO is not convincing and does not correspond to the standard established by the Schiebel group. As suggested above, the authors should perform this experiment in a different cdc5 mutant that is fully active under permissive conditions.

We have done our best to optimize the conditions to better visualize the mobility shift of the Spc72 band in Western blots after inhibition of Cdc5 activity and we believe that we now more unambiguously demonstrate that addition of the CMK-C1 inhibitor leads to disappearance of the slowest migrating forms of the γ-TuCR (new Figure 1H). To facilitate visualization of the results, we further included intensity plot of the lanes in the Western blots, as previously used in prior studies of Spc72 phosphorylation (Maekawa et al., *eLife* 2017) (new Figure 1I). These assays allowed us to demonstrate that Spc72-GFP displayed the same electrophoretic mobility both in wild type cells and in the untreated *cdc5-as1* mutant, ruling out the concern raised by the reviewer (new Figures 1H, I). Furthermore, and remarkably, Spc72-GFP showed a similar electrophoretic migration pattern in *cdc5-1* cells after CMK-C1 addition than that displayed by the Spc72-AA mutant protein in untreated wild type cells (new Figure 1H, I).

The Abstract of the manuscript does not describe well the results of the study as it contains very few pieces of new information (the last sentence only). It would be to the benefit of this study if the authors would include more information about their important results in the Abstract.

The Abstract has been modified. Hopefully now it does better describe the relevance of our results.

Reviewer #2:(Comments for the author)Taken together this is an interesting study but the authors proposals go far beyond what might be permitted by the data presented without further development. It is important to establish if inactivation of Cdc5 using cdc5-as causes first cycle arrest or not, to validate the authors' position regarding carry over effects on subsequent cell cycles. The relationship between Cdc5, Spc72 and Kar9 should be better understood in molecular terms for true advancement here given the body of knowledge on these proteins. The interaction between Cdc5 and Spc72 has been studied mainly in the context of MEN-SPOC with multiple connections suggested. Furthermore, others have previously implicated components of the MEN directly in control of SPB inheritance via Kar9 (Y Barral, S Piatti). It follows that the mechanistic involvement of Cdc5 here is far from demonstrated.

We thank the reviewer for considering our analyses of interest. To hopefully satisfy her/his concerns, we have made a significant effort to reinforce our data so that our conclusions are further supported by the new evidences provided. Regarding the doubt raised with respect to *cdc5-as1*, it is important to first clarify that this allele is indeed able to cause a first cycle arrest (Figure 3A). Additionally, and as extensively discussed in our response to reviewer #1, our new results with additional *cdc5* mutant alleles further discard now that our initial observations could be associated to an allele-specific defect (new Supplementary figure 1C and Figures 1G-I and 2E-H). Moreover, the use of the *cdc5-16* allele, which encodes a Cdc5 mutant protein that does not load on the SPBs, as well as that of the mutant Spc72-AA (S231A, S232A) protein provided us with the means as to add further mechanistic insights into the role of the POLO-like kinase in the asymmetric inheritance of the SPBs in budding yeast. Finally, we beg to differ with the reviewer in that the role of Cdc5 could be either inferred from previous analyses of the control of SPB by the MEN pathway or be necessarily restricted to the MEN-SPOC interaction in the context of the SPB. In this sense, not only we demonstrate that, in contrast to the MEN, Cdc5 has multiple roles in SPB inheritance, but we further provide evidences that it acts both on the SPBs (controlling Spc72 localization) and outside these structures (regulating Kar9).

One additional puzzle not addressed by the authors is the prospect of Kar9 being enriched in the SPB retained by the mother cell. Normally Kar9 build-up marks the pole destined to the bud irrespective of SPB age distinction. The authors report here that Kar9 localisation was "reversed". Can the authors explain why the pole recruiting Kar9 more heavily is not delivered to the bud? The authors should provide live imaging analysis to evaluate Kar9 dynamic function upon Cdc5 inhibition in order to understand the basis for this apparent defect. The images provided convey little meaningful information as, on each case, a single cell is shown with label presumably at the end of an astral microtubule. A detailed characterisation of modes of Kar9 label (pole, microtubule length and microtubule plus ends) should have been provided as the pattern of Kar9 distribution is in fact more complex than what is presented here. Finally, additional characterisation should be implemented through cell biology to refine the viewpoint that inactivation of Cdc5 holds consequences in subsequent cell cycles. Perhaps, a dose-response to cdc5-as inhibition might be attempted to fully document the behaviour of this allele, to begin with. It may also be important to establish the kinetics of Cdc5 inactivation by the inhibitor in their hands in order to entertain the various scenarios contemplated in the manuscript.

The possibility that Kar9 association with the SPBs gets reversed while the pole recruiting more Kar9 is retained in the mother cell and not delivered to the daughter could be explained if binding of Kar9 to Myo2 was somehow defective, so that the SPBs would be pulled towards the bud in a Dyn1-dependent manner. This, in fact, would agree with previously published data that suggest that blocking Kar9-Myo2 interaction does not impair Kar9 loading on the SPBs (Lengefeld et al., 2018). In any case, and as suggested by the reviewer, we have re-evaluated Kar9 localization more carefully, this time analyzing the global distribution of Kar9 to all cellular locations. As previously indicated to reviewer #1, this new analysis allowed us to uncover a new remarkable phenotype after the inactivation of the POLO-like kinase. Our data show that inhibition of Cdc5 led to problems both in the mobilization of Kar9 from the nucleus to the cytoplasm, with the protein displaying increased interpolar microtubule localization, as well as with the establishment of the polarized distribution of Kar9 towards the daughter cell (new Figures 2E, F). This, together with our results that indicate that Cdc5 loading on the SPBs is not necessary to regulate Kar9 function on SPB inheritance (new Figure 2E), has important implications in terms of the mechanism that we proposed for the role of Cdc5 in this process, as most extensively detailed in our answer to point #12 and as also previously explained in our response to reviewer #1. We do thank the reviewer for her/his suggestion.

Finally, it is important to note that we did indeed optimize the concentration of the CMK-C1 inhibitor to be used in our experiments. Our results show that at lower inhibitor doses than the one used in our initial manuscript cells did not arrest, while higher concentrations led to similar randomization levels of SPB fate. Since results were similar with higher doses, we decided to maintain the minimal concentration that facilitated the maximum inhibition of Cdc5.

The authors should also provide a more complete account of the microscopy analysis performed throughout. In the Materials and methods, a list of antibodies is mentioned in connection with western blot analysis and immunofluorescence. It was not made clear what part of the microscopy analysis was based on imaging fluorescent tags (whether live or fixed?) or by immunofluorescence. In addition, the protocols for quantitative imaging analysis should be described in full.

The text has been modified, as requested by the reviewer.

Specific comments:1) First paragraph of Results section: yeast SPB duplication is "conservative" and not semiconservative as stated there. This is why the slow-folding tag discriminates SPBs by age. Please correct the text.

Yeast SPB duplication is sometimes referred to as “semi-conservative” because, despite most of their structural components being mainly incorporated “de novo” in the newly generated SPB, some display a higher exchange rate between the pre-existent and the new SPBs (Lengefeld et al., 2018). Since, in order to discriminate SPBs by age, the slow-folding fluorescent tag is introduced in one of the conservatively duplicated SPB components, we have modified the text according to the reviewer’s suggestion to avoid possible confusions.

2) Regarding localization of Cdc5 relative to SPB duplication – how do the authors establish the timing of SPB duplication by widefield microscopy?

In our analyses, we simultaneously localized Cdc5 and Sc72. We thus used Spc72-mCherry itself as a SPB marker since, despite being asymmetrically localized, it can be observed on both spindle poles, as already indicated in our original manuscript.

3) Cdc5-GFP labels a number of structures making the quantitative evaluation of spindle pole labels difficult. How was the quantification performed here? Could you provide sample images along with details on the analysis? Are cells scored visually or symmetry/asymmetry determined through measures?

We have scored Cdc5 localization visually, and we have restricted our observations to its distribution to the SPBs, which can be clearly observed despite the POLO-like kinase being also present in other cellular locations, as correctly indicated by the reviewer. To illustrate the distribution of Cdc5 in our cellular background we have provided sample images, as requested (new Supplementary figure 2A).

4) Regarding the discrepancy between findings in Figures 1 and 3 on the extent of randomised inheritance and Spc72 bias. Did the authors establish the extent (if any) of loss of bias as a result of impaired Cdc5 when entering subsequent cell cycles? Is the bias to the old SPB ever lost? What is the corresponding impact on Kar9 localisation AND function?

We have evaluated the association of Spc72 with the old SPB during the second division after *cdc5-as1 tab6-1* cells synchronized in G1 were allowed to enter the cell cycle in the absence or in the presence of CMK-C1 inhibitor. We found some problems while doing these experiments since, in order to follow the *tab6-1* allele during the strain crosses, we had to tag it with a different epitope that the one used in our original experiment. For unknown reasons, tagging Tab6-1 with 13Myc rather than with yEGFP, despite still allowing *cdc5-as1 tab6-1-13Myc* cells to enter a second cell cycle in the absence of POLO activity, did prevent them to carry out cytokinesis. In any case, and since we could still distinguish the daughter from the mother cell during the subsequent division, we analyzed Spc72 and Kar9 distribution as requested by the reviewer. This analysis indicated that the preferential bias that Spc72 showed towards the pre-existent SPB was still maintained as a result of the inhibition of Cdc5 activity when cells entered a second cell cycle, in agreement with loading of new γ-TuCR being dependent on Cdc5 activity and localization to the spindle MTOCs (new Supplementary figure 5C). Our new data also shows that the bias towards the old SPB was also maintained during the first cell cycle for the Kar9 protein (new Supplementary figures 6A, B). Additionally, the experiments using the *cdc5-16* allele encoding a mutant kinase that cannot load to the SPBs further indicate that Cdc5 exerts its control over Kar9 outside these structures (new Figures 2F, H). However, and excitingly, Kar9 preferential association to the pre-existent SPB is affected in *cdc5-as1 tab6-1* cells that entered a second cell cycle after being released from an initial G1 arrest in the presence of CMK-C1 inhibitor (new Supplementary figure 5D). These results, as a whole, support our initial hypothesis that Cdc5 facilitates the establishment of SPB identity during mitosis, thereby ensuring that the asymmetric segregation of the preexistent SPB can be maintained during the subsequent cell cycle, and that this is likely mediated by modification of Spc72 in a manner that facilitates preferential association of Kar9 to this γ-TuCR during the following cell cycle.

5) According to Figure 4. inactivation of Cdc5 prevented Spc72 symmetry increase while Cdc5-Spc72 interaction displayed a reduced bias to the daughter-bound SPB. How do you reconcile these observations? Should there be a bias to either pole?

To answer to this apparent conundrum, it is important to highlight again the fact that we support that Cdc5 plays multiple roles in the establishment of the SPB inheritance pattern. Cdc5 inactivation precludes loading of new Spc72 protein both on the pre-existent and the newlygenerated SPBs, therefore preventing the final symmetric localization of the γ-TUCR in anaphase. However, our data indicate that Cdc5 activity is not required for its interaction with Spc72. The reduced bias to the daughter-bound SPB in the Cdc5-Spc72 interaction is consistent with our observations, since their association would be initially favored in the context of the old SPB, which is the one carrying more γ-TUCR protein when the cell cycle starts. Then, the lack of Cdc5 activity would make it impossible for their association to be modulated, and the fate of this SPB would be later randomized as a consequence of the additional perturbation of Kar9 function due to POLO kinase inactivation.

6) In Figure 5, it is unclear why the profiles in B might be different within the auxin-treated window with DMSO vs no CMK-C1? Could you indicate exactly when the inhibitor was added and clarify?

As pointed out by the reviewer, and as discussed previously in one of reviewer #1’s minor comments, the differences in the profiles in Figure 5B are likely due to the CMK-C1 inhibitor being added 15 min before cells were released from the metaphase arrest (*i.e.*, at 75 min). We do apologize again because this is a missed piece of information that was necessary to properly evaluate the results. We have subsequently also corrected it in the graph and the legend of the figure.

7) The authors claim that SPBs intrinsically promote Kar9 bias but quote Lengefeld et al., 2018 without elaborating that the position of that study is in fact at odds with this claim. This should be discussed correctly for any points supporting and contradicting the author's proposal here regarding the contributions by the intrinsic asymmetry of SPBs.

The reviewer is right in that the way that the statement was written leads to confusion. Lengefeld et al. postulated that it is not the kinetics of SPB maturation that controls the asymmetry of astral microtubule organization between the preexisting and new SPBs. Instead, it is the regulation of microtubule dynamics and γ-TUC function on both SPBs, as well as the control of the recruitment and maintenance of Kar9 on the preexisting SPB, that ensure the pre-established SPB inheritance pattern (Lengefeld et al., 2018). This agrees with our results showing that Cdc5 respectively controls both Spc72 and Kar9 function. Regrettably, and although this is what we meant to underline in the Discussion of our initial manuscript, it is definitely true that when we used the term “intrinsic nature of the SPBs” to refer to modifications of these factors taking place on the SPBs, we were not correctly reflecting the idea transmitted by Lengefeld et al., but in fact getting at odds with their data. Furthermore, our new results make this point even more evident when we demonstrate now that Cdc5 does not need to load on the SPBs to regulate Kar9 function. Therefore, we have subsequently removed the initial statement and modified it as follows: “The evidences shown here support the idea that it is not the kinetics of SPB maturation, but fundamentally the regulation first of microtubule dynamics and γ-TUC loading on both SPBs and then of Kar9 recruitment and maintenance to these structures that ensure the pre-established SPB inheritance pattern (Lengefeld et al., 2018)”.

8) At various points the authors conclude functional interactions on the basis of localisation and attempt to infer mechanistic links between Kar9, Spc72 and Cdc5. Such claims remain speculative at best in the absence of concrete biochemical data in the current study.

We have incorporated new data to hopefully strengthen other aspects of our characterization besides the localization analyses. In this way, and together with our initial Western blot analyses and BiFC assays (which can be considered also as a biochemical association assay), we have further expanded the evaluation of the biochemical interactions between Cdc5, Spc72 and Kar9. As such, we have improved the resolution of the Western blot analyses, including the effect of POLO-like kinase inactivation on both Spc72 and Kar9 post-translational modifications by using different mutant alleles of *CDC5* (new Figures 1H, I and 2H). Moreover, as previously indicated in our response to reviewer #1, we have generated a Spc72 mutant lacking two serine residues that are bound by Cdc5 (S231A, S232A) (Snead et al., 2007). This Spc72-AA mutant showed randomized distribution (new Figure 1G) and reduced levels of loading on both SPBs (new Figure 3F), as observed for wild type Spc72 after Cdc5 inactivation. However, SPB fate and Kar9 preferential distribution to the old SPB were not affected in the *spc72-AA* background (new Figures 1J, 2F), in agreement with what found for mutants expressing a Cdc5 kinase that cannot load on the SPBs. Overall, we believe that these new results do provide additional support to our initial data and definitely reinforce our conclusions from a biochemical perspective.

9) While not immediately relevant to the manuscript, the Discussion on the relationship between CDK and Kar9 recruitment relies on a single study. This should be correctly revised for the most coherent view by considering collectively studies from Barral, Miller and Schiebel laboratories.

According to the reviewer’s suggestion, we have included the references to the work of the different laboratories that have contributed to our understanding of the role of CDK in the regulation of Kar9 localization on the SPBs. However, due to the amount of new information that we had to add during the revision and since this topic, as in fact pointed out by the reviewer, is not immediately relevant to our manuscript, we have just limited to indicate in the text that CDK regulates Kar9 distribution on the SPBs, without getting into further details.

10) How do the authors interpret the cell cycle-dependent pattern of phosphorylation of Kar9 upon Cdc5 inactivation shown in Supplementary figure 1?

Kar9 is a highly post-translationally modified protein, which is phosphorylated, ubiquitinylated, sumoylated and regulated by many different factors (Leisner et al., 2008; Kammerer et al., 2010; Hotz et al. Cell 2012). The data from Supplementary figure 1G shows that, despite clearly affecting the post-translational modification profile of Kar9 throughout the cell cycle, Cdc5 inactivation does not fully eliminate all modified forms of the protein, which accumulates in an intermediate modified form when cells finally arrest in anaphase.

11) Supplementary figure 2 E-H is not incorporated in the narration. Could the authors provide a direct account summary of their findings for pattern of inheritance, Spc72 and Kar9 bias upon Cdc5 inactivation in asynchronous cells versus cells released from α factor?

Supplementary figures 2E-H were indeed referenced and correspond to Spc110dsRed distribution (SPB inheritance), not to Spc72 or Kar9 bias to the old and new SPBs. As indicated in our initial manuscript, these figures show that the asymmetric pattern of SPB inheritance was maintained after cells were released from a G1 arrest and synchronously progressed through the cell cycle in the absence of Cdc5 activity. In summary, our results in exponentially growing cells indicate that asymmetric SPB inheritance is randomized after Cdc5 inactivation. This randomization results from:

– Defects in Spc72 loading, but not in its association with the pre-existent SPB.

– Problems in the mobilization of Kar9 from the nucleus and also with its association with the preexistent SPB, which can be observed due to cells being exponentially dividing when the inhibitor is added.

Finally, and regarding cells synchronously entering the cell cycle in the absence of Cdc5 activity, SPB inheritance is not affected during the first cycle, but it is then disrupted during the subsequent cell cycles if cells are allowed to escape from the anaphase arrest induced by the lack of POLO activity. Under these conditions:

– Spc72 remains preferentially associated to the pre-existent SPB in every cell cycle, since loading of new protein is severely affected in the absence of Cdc5 activity.

– Kar9 associates to the old SPB during the initial cell cycle, facilitating the normal pattern of SPB inheritance during this division. However, Kar9 fails to properly recognize the pre-existent SPB during the second cell division, explaining the disruption of the predetermined SPB fate. – The previous results suggest a failure to properly establish SPB identity, likely by a defective Cdc5-dependent Spc72 modification that interferes with its normal association with Kar9.

12) What is the authors' proposal for the actual path for the reversal of the pattern of inheritance? Is it due to SPB symmetry, Kar9 disfunction or both? It should be possible to demonstrate their proposal using live imaging data at least.

We are sorry that we were not able to more clearly explain our model for the role of Cdc5 in establishing SPB fate during budding yeast mitosis. We tried to clarify it, further including the new mechanistic aspects of the process uncovered during the revision, and hopefully it is now easier to follow. As explained in our answer to the previous comments from the reviewer, and in brief, our observations agree with a model according to which Cdc5 would facilitate the establishment of the asymmetric SPB inheritance pattern by carrying out at least three different functions:

i) To prevent loading of new Spc72 protein on the SPBs until the metaphase-to-anaphase transition, therefore favoring the microtubule nucleation capacity of the pre-existing SPB, which is the one that carries Spc72 from the previous cell cycle, and its anchoring to the daughter cell cortex.

ii) To facilitate the preferential asymmetric distribution of Kar9. In contrast to the control of Spc72 loading on SPBs, this activity of Cdc5 does not require localization of the kinase to these structures. We favor the idea that Cdc5 could act preventing a correct post-translational modification of Kar9 in the nucleus, impairing both its capacity to be mobilized from this cellular compartment and to associate with the pre-existent SPB and polarize towards the daughter cell. iii) To confer the SPB its identity in the prior cell cycle so that it is recognized as the old SPB in the subsequent division. We now further support the idea that this identity mark is imprinted at least in part by a Cdc5-dependent modification of Spc72, which facilitates preferential binding of Kar9 to this structure.

These different functions turn the POLO-like kinase Cdc5 into a “molecular timer” that facilitates the proper segregation of the pre-existent SPB into the daughter cell.

13) In the context of return to growth from a nocodazole block, what happens to Kar9 upon Cdc5 inactivation?

In agreement with our observations suggesting that the identity mark for the old SPB is established during the prior cell cycle, in the context of return to growth from a nocodazole block Kar9 still preferentially associates to the pre-existent SPB (new Supplementary figures 6C, D), similarly to what observed during a single cell cycle. As predicted by reviewer #1 in one of his/her comments, and according to our hypothesis that Cdc5 enables Spc72 loading on both SPBs at the metaphase-to-anaphase transition, the fact that SPBs are randomized after cells are allowed to exit the metaphase block and repolymerize the mitotic spindle can be explained due to the fact that Spc72 gets prematurely loaded symmetrically on both SPBs providing them with a more similar microtubule nucleation capacity.

14) In most synchrony experiments the possible effect of Cdc5 inactivation on progression of the spindle pathway is unapparent. Previous studies point to impaired progression from short to elongated spindles. Have the authors verified that the inactivation of Cdc5 in their study is absolute?

The inactivation of Cdc5 with CMK-C1 is indeed effective, as demonstrated by the final anaphase arrest displayed by the cells and the post-translational modification defects shown by Spc72 and Kar9. We did notice somewhat perturbed spindle microtubule morphology, but this did not preclude cells to reach the previously mentioned anaphase block. Finally, and more importantly, we now demonstrate that we obtained similar results by using different *cdc5* alleles, further reinforcing the validity of our conclusions.

15) According to their model, what might be the maximal extent of "reversed" inheritance on a given cell cycle? Has this been quantitatively correlated with Spc72 or Kar9 distribution?

The extent of reversed SPB inheritance varies depending on the strain background and growth conditions, with maximum percentages of retention of the old SPB within the mother cell on a given cell cycle ranging from 5 to 10%. The percentages that we obtained are similar to those previously described by other groups (*e.g.*, averages of 2% (Pereira et al., 2001) or 5% (Hotz et al., Cell, 2012), among others). Furthermore, the mother:daughter cell ratio in terms of SPB inheritance observed in exponentially growing cells after Cdc5 inactivation were similar to what previously described for other mutants that randomize SPB inheritance, including that reported for a *kar9*Δ mutant in the original paper in which this phenomenon was first described (30:61 ratio; Pereira et al., 2001) and others estimated later by different groups (*e.g.*, 34% cells with old SPB retained in the mother cell of a *kar9*Δ mutant; Hotz et al. Cell, 2012).

On the other hand, and regarding Spc72 and Kar9 localization, it is widely established that Kar9 distribution to the SPBs is more asymmetric than that of Spc72, which despite preferentially loading on the pre-existing SPB is more evidently observed also in the newly generated one (Juanes et al., 2013). As such, Spc72 was estimated to be asymmetrically distributed in approximately 70% of wild type metaphase cells, to then become symmetrically localized in close to 75% of anaphase cells (Juanes et al., 2013). These levels are comparable to those obtained in our analyses of Spc72 distribution. However, Kar9 has been reported to asymmetrically localize in a higher percentage during metaphase (*e.g.* averages of approximately 85% (Hotz et al. Cell, 2012; Lengefeld et al., 2017), 90% (Meednu et al. Genetics 2008; Cepeda-García et al. Mol Biol Cell 2010) cells with Kar9 distributed towards the daughter cells) and to remain asymmetrically localize during anaphase (*e.g.* averages of 90% (Liakopoulos et al., 2003; Cepeda-García et al. Mol Biol Cell 2010), 99% (Meednu et al. Genetics 2008) cells displaying preferential Kar9 distribution in the bud). Again, these numbers are similar to the percentages estimated in our experiments.

Besides the previous analyses, it is important to highlight that a further simultaneous evaluation of Spc72 and Kar9 localization during the cell cycle showed that, when both proteins were asymmetrically distributed, they displayed a coincidental localization to the SPBs in up to 80% of anaphase cells, which is consistent with the previously indicated percentages. Moreover, this analysis also allowed us to demonstrate for the first time that Kar9 localization to the SPBs depends on Spc72 presence on these structures (Figure 7E).

Finally, the results obtained during the revision of our manuscript indicate that while both Cdc5 activity and loading to the SPBs are required to facilitate the correct loading of Spc72 on the SPBs, only the activity of the POLO-like kinase is needed in order to support the preferential association of Kar9 to the old SPB and maintain the pre-determined SPB inheritance pattern, thus allowing us to conclude that loss of Cdc5-dependent regulation of Kar9 has more detrimental consequences on the determination of SPB fate than that of Spc72.

Therefore, and as a whole, we consider that our data concerning SPB fate as well as Spc72 and Kar9 distribution is consistent, coherent and comparable to previous observations by other laboratories, further providing new relevant information about the dynamics and regulation of these processes.

[Editors’ note: Reviews of the revised manuscript from another journal follow.]

Reviewer #2: (Comments for the author)Comments on revision of manuscript by Matellán et al.I have considered the revised manuscript along with the detailed response to the reviewer´s initial comments. Part of the comments were addressed by revisions with additional data provided. The authors make a strong case for the impact of Cdc5 inactivation on Spc72 recruitment, on the one hand, and on the pattern of SPB inheritance on the other. These are interesting observations worthy of publication. However, the nature of the corresponding mechanistic connection as well as the involvement of Kar9 remains elusive without directly addressing the ensuing effect on microtubule dynamics and Kar9 activity beyond static images. Cdc5 direct effects on astral microtubule distribution and function in line with impairment of Spc72 were not explored here.

We thank the reviewer for acknowledging our effort to provide a detailed response to all her/his initial comments. We made a meticulous, thorough and exhaustive effort to answer each and one of the reviewer’s concerns in what we believe was a rigorous manner, and hence we regret that we were yet unable to finally obtain a more positive outcome. Nonetheless, we still consider that we could provide the reviewer with an answer for her/his remaining concerns, including some additional observations and alternative experimental approaches that could hopefully make her/him to be supportive of the final publication of our manuscript.

The main concern still raised by the reviewer is the lack of dynamic data. We do not however fully agree with the reviewer’s statement that we did not supply data to evaluate the impact of the lack of Cdc5 activity on the dynamics of Kar9 and Spc72. Although it is true that we could not finally incorporate live cell imaging videos, which the reviewer suggested as a possible option to answer one of the 15 points raised during her/his initial revision of the manuscript (point #12), many of our analyses of cell cycle progression indeed allowed us to evaluate the dynamics of the proteins on the SPBs by assessing their localization in subsequent time points. We did in fact try to follow the dynamics of the proteins using real live imaging microscopy but we faced many experimental difficulties that, given the limited time for the initial review, prevented us to finally provide videos of the process. Specifically, the main problems were the following:

– The CMK-C1 inhibitor does not work properly in minimal medium and the rich medium used for the experiments described in the manuscript has a high auto-fluorescence that made it extremely hard to visualize Spc72 and Kar9 signal.

– We could not use the thermosensitive mutant alleles of *CDC5* either, since for some reason when changing the cells from the liquid culture to the incubation plates at the restrictive temperature in the microscope the cells stopped dividing and did not progress into the cell cycle.

In order to satisfy the reviewer’s concern, we believe that we can provide instead alternative new evidences that Cdc5 has a direct effect on modifying the activity of Spc72 on the SPBs and subsequently the capacity of these structures to nucleate microtubules. Specifically, after we submitted the revised version, we have evaluated the distribution of the γ-tubulin Tub4 and we have observed that, as for Spc72 distribution, the lack of Cdc5 activity causes a randomization of the loading of Tub4 on the SPBs. This is a strong evidence of a direct effect on the functionality of Polo-like kinase on the nucleation capacity of Spc72. We have done the experiment once, and we would have just to repeat it to carry out the statistical analysis. On the other hand, we could finally add further support to our hypothesis by evaluating the effect on the lack of Cdc5 activity on the number and distribution of the array of astral microtubules. Since we have the strains ready, this experiment could be also carried out in a in a reasonable time (three to four weeks). In summary, we strongly believe that all the cumulative data provided during the revision process, as already acknowledged by reviewer #1, together with these new proposed approaches will clearly establish the direct mechanistic connection between Cdc5, Spc72 and the microtubule nucleation capacity of the SPBs requested by reviewer #2. Hence, we urge the reviewer to consider, based on the abovementioned technical difficulties to obtain live cell imaging videos, the alternative proposed experiments together with all our previous evidences to be sufficient as to satisfy her/his concern.

The experiment involving nocodazole wash out is also poorly depicted as Kar9 is known to break symmetry while eliciting a random pattern of inheritance irrespective of Spc72.

We are sorry because this issue is probably just something that we did not sufficiently explain in our manuscript and could be thus easily solved by rewriting parts of the text in order to clarify it. We beg to disagree in that the experiment is poorly depicted. Moreover, it fully fits our predictions and the impact of the lack of Cdc5 activity on Kar9 is indeed evaluated. In this sense, it is important to emphasize that despite the reviewer is right regarding his comment about Kar9, we demonstrate in our manuscript for the first time that localization of Kar9 depends on that of Spc72 and that their interaction is likely regulated by Cdc5. Finally, our results show that Kar9 still follows the SPB that enters the daughter cell after the nocodazole treatment in the absence of Polo-kinase activity, as expected according to our hypothesis.

Another point pending from the previous comments is that while the authors studied alternative cdc5 alleles (demonstrating that the observations are not solely due to cdc5-as) the question of the discrepancy between asynchronized and synchronised cells entering the cdc5-as block remains unexplained given the assertion that cells experience first cycle arrest. Furthermore, there seems to be an internal discrepancy in Figure 1 regarding the behavior of Spc72AA (compare Figure 1G and 1J) with regard to whether inheritance is lost or not.

We did provide an explanation for the discrepancy between asynchronized and synchronous cultures in our manuscript. As discussed, and in contrast to what happens in cells released from a G1 arrest that enter the cell cycle already without Cdc5 activity, inactivation of Cdc5 in an asynchronous exponential culture provides a temporal window during which cells at different cell cycle stages progress through mitosis with reduced Polo-like kinase activity until eventually reaching a final anaphase arrest after its complete inhibition. This transient partial inactivation of Cdc5, together with the fact that the Polo-like kinase further regulates additional downstream events in the determination of SPB inheritance besides specifically controlling Spc72 association with the SPBs at the metaphase-to-anaphase transition (*e.g*., the interaction of Kar9 with the pre-existing SPB), would explain the randomized SPB fate observed in this situation.

On the other hand, and regarding Figures 1G and 1J, we do not find any discrepancies here. Although Spc72AA is randomized (Figure 1G), Cdc5 activity can still correctly regulate Kar9 function and therefore, similarly to what found for the *cdc15-16* mutant (Figure 1G, 2F, 2G and Supplementary figure 1C), the final asymmetric destination of the SPB is only slightly affected (Figure 1J). This was indeed already highlighted in our revised manuscript, in which we discussed how Cdc5’s role on regulating Kar9 activity seems to be more important for SPB inheritance determination than that on Spc72 function.

As I indicated before, the data suggest that Cdc5 inactivation perturbs Kar9 function in astral microtubule delivery from one SPB as well, since Kar9 localisation is "reversed" without due correlation to bud-fate. The authors concede in the accompanying letter that Kar9-mediated transport, for example, might be affected. Yet, this possibility is not explicit in the manuscript. This apparent effect of Cdc5 could be the primary determinant for disruption of SPB inheritance indirectly and irrespective of Spc72. An observation that should be fully developed when attempting to link Kar9 modification, Cdc5 and inheritance.

As indicated before, we do agree in that the role of Cdc5 on Kar9 activity seems to be more important than that exerted on Spc72. In fact, and as also mentioned, we did explicitly include this observation in our revised manuscript. We will nonetheless further develop this aspect in a new version of our manuscript following the reviewer’s suggestion. Despite this, we would like to emphasize that our results clearly demonstrate that Cdc5 regulates both Kar9 and Spc72. Moreover, it is extremely important to highlight that, for the first time, we demonstrate that the localization of Kar9 to the SPBs is fully dependent on Spc72 and that this is likely regulated by Cdc5, which further strengthen the importance of the interconnection between the three proteins for the correct establishment of the SPB inheritance pattern.

Finally, I found the revised manuscript very hard to read at various points, given the trend for circular arguments underlying the presentation of the data. Even the revised summary fails to indicate in plain terms what the study actually shows. In this regard, I further feel that the manuscript is not ready in its current form as it fails to link robustly data and interpretation.

We will edit the manuscript and simplify the writing to hopefully more precisely link data and interpretation. We will pay special attention to the Abstract, so that it summarizes in a more straightforward manner the main conclusions drawn from our article.

[Editors’ note: Reviews of an appeal from another journal follow.]

Reviewer #2:

We want to thank the reviewer for her/his answer to our appeal after the editorial decision, as well as to apologize in case she/he got the impression that we were reluctant to implement her/his original suggestion regarding the live-cell imaging analyses. In our rebuttal letter, we just simply wanted to indicate that we were finally unable to carry out these experiments, to explain the reasons that justified our limitations to successfully use this approach and to argue why we believed that the additional experiments that we provided were indeed a valid alternative to support our conclusions. We want to further acknowledge the reviewer for providing us with detailed suggestions to strengthen our manuscript, taken that we could eventually satisfy her/his concerns. In this sense, we would like to highlight that we have made a substantial effort to carry out all the proposed experiments, including even some that the reviewer considered not strictly necessary but still highly relevant to directly prove that Cdc5 works through a mechanism linking both Spc72 and Kar9. Moreover, we have included novel additional experimental evidences to more strongly corroborate our hypotheses. Please, find below our response to all the points raised, which we believed that we have managed to answer in a satisfactory manner.

Their data supports the idea that Cdc5 is required for correct SPB inheritance. However, that this is controlled via Spc72 is not demonstrated: I requested that the path of events over consecutive cell cycles linking strictly Spc72 behaviour and SPB inheritance was depicted throughout – This was not done. After all, synchrony experiments indicate that, for sure, Cdc5 action is not “first cycle” – visualising the correlation over 2 or more cell cycles would have been necessary for proof.

Our manuscript does demonstrate that Cdc5 is required for correct spindle pole body (SPB) inheritance in *Saccharomyces cerevisiae*. As the reviewer points out, an important role that we have uncovered for Cdc5 in this process is mediated by the control of Spc72 activity. Specifically, Cdc5 regulation of this ɣ-tubulin complex receptor (ɣ-TuCR) specifies SPB fate by promoting the preferential nucleation of microtubules emanating from this structure during the initial stages of mitosis, thus facilitating the distribution of the old SPB towards the daughter cell. Accordingly, we now provide new data demonstrating that distribution of ɣ-tubulin (Tub4) on the SPBs, which preferentially accumulates on the daughter-destined SPB up to the metaphase-to-anaphase transition in an unperturbed cell cycle (Juanes et al., 2013), is randomized when Cdc5 kinase activity was inhibited (Supplementary figure 6D), strongly supporting our initial conclusions.

The role of Cdc5 in the establishment of the asymmetric SPB fate in budding yeast is however not only dependent on Spc72. This is sustained by the fact that neither Cdc5-16, which does not load on SPBs, nor Spc72-AA, which is not phosphorylated by the Polo-like kinase, affect the SPB inheritance pattern despite Spc72 being randomly distributed in both cases. Indeed, in addition to its activity on Spc72, we propose that Cdc5 has a more direct role in determining SPB fate that is mediated by the control of Kar9 function and that does not need loading of the Polo-like kinase on the SPBs. We instead suggest that this regulation likely takes place in the nucleus, as supported by our experimental data. Accordingly, even though distribution of Kar9 on the SPBs is randomized when Cdc5 activity is inhibited (Figure 2B), its localization to these structures is not affected in the *cdc5-16* or *spc72-AA* mutants (Supplementary figures 2B, 2C). We agree with the reviewer, nonetheless, in that the link between Kar9 and Cdc5 is the most unexplored aspect of our manuscript. Consequently, we have made an additional effort to shed further light on this particular issue to hopefully clarify the functional connection between these proteins (see below in our response to the rest of the reviewer’s concerns).

Finally, as the reviewer indicates, our experiments with synchronized cell cultures also support that Cdc5 confers the SPBs their identity, but that this is only evident in the following cell cycle. In order to provide further experimental support to this idea, we have designed an alternative new approach that we believe strongly sustains our claim and that will hopefully contribute to satisfy the reviewer’s concern. Specifically, we have evaluated SPB inheritance in a strain background in which we combined both the *cdc5-as1* and the *spc72-AA* mutant alleles. In *cdc5-as1spc72-AA* cells growing in the absence of Cdc5 inhibitor, the γTuCR cannot be phosphorylated by the POLO kinase on the SPBs and the fate of these structures cannot be established during the current cell cycle, so that the initial scenario would be similar to that of *tab6-1 cdc5-as1* cells that have gone through a “first cycle” of Cdc5 inhibition. As a consequence, after a

pheromone-induced arrest of *cdc5-as1spc72-AA* cells in G1 and their subsequent synchronous release in conditions that inhibit Cdc5 kinase activity, we should observe a synergistic effect of the inactivation of the Polo-like kinase with the Spc72-AA mutation, resembling what happens during a “second cell cycle” in our synchronization experiments with *tab6-1cdc5-as1* cells. Excitingly, in agreement with our hypothesis, while the asymmetric SPB distribution was still preserved when Cdc5-as1 was not inhibited, SPB fate was notably affected as a consequence of the lack of POLO activity in the *spc72-AA* background (Supplementary figure 6C). This result, together with all the previous evidences, are strong pillars to sustain a role of Cdc5 in conferring the old SPB its fate during the previous cell cycle.

Separately, Kar9 function upon Cdc5 inactivation may be perturbed. In wild type cells it is known that there is an absolute correlation between Kar9 bias and SPB bud-ward fate, as it is Kar9 that delivers the pole. Here they showed that upon Cdc5 inactivation, Kar9 often favours the SPB remaining in the mother cell i.e. presumably Kar9 function in bud-ward delivery has been disrupted while localisation to SPBs is retained – and not particularly correlated with Spc72. Because Kar9 recruitment depends partly on Spc72 (others have shown the involvement of Stu2, Tub4, Bim1, etc., all Cdc5 substrates), the authors claim that the two targets are unified in a single mechanism. They offer no proof of this. However, alluding to the number of points originally raised in my review is no substitute for this deficiency. I haven’t requested biochemical data in my original review, having settled for correct functional understanding of the underlying link between Kar9 and Cdc5 through imaging. However, if the authors were to insist on their position, then, in vitro demonstration that Cdc5 acts directly to promote Spc72-Kar9 interaction would be necessary to support this idea. The in vivo tag complementation only demonstrates an overall link between Spc72 and Cdc5. That specific connection is not new. I also suggested the alternative to focus the current manuscript solely on Spc72 and Cdc5 giving the authors the opportunity to gather the appropriate proof for Kar9 role for a future publication.

Although the reviewer is right in that prior studies have shown the involvement of other proteins in the loading of Kar9 on the SPBs, our results demonstrate for the first time that Kar9 recruitment to these structures does fully (not partly) depend on Spc72. Importantly, now we additionally show that lack of Kar9 binding to the SPBs in the absence of Spc72 is not due to protein degradation (Figure 7G). Furthermore, in agreement with our results with the *cdc5-16* and the Spc72-AA mutants (Figure 2H, Supplementary figures 2B, 2C), Kar9 was still post-translationally modified when Spc72 was not expressed (Figure 7G), which suggests that Cdc5 facilitates Kar9 asymmetric distribution by acting on this protein in the nucleus, likely impairing its capacity to be mobilized from this cellular compartment, to associate with the pre-existent SPB and to polarize towards the daughter cell.

We do believe that this data is new, highly relevant to understand the mechanisms regulating spindle orientation and SPB inheritance, and overall a proof of a novel direct functional connection between Cdc5, Spc72 and Kar9. Nonetheless, we agree with the reviewer in that including biochemical data that further reinforce our arguments would be a strong additional evidence to more categorically establish this link. As such, and despite the reviewer did not specifically ask for it, we evaluated the capacity of Spc72 and Kar9 to co-immunoprecipitate with each other, as well as the potential dependence of this association on Cdc5 activity. Remarkably, according to our hypothesis, these analyses demonstrate that Kar9-13Myc coimmunoprecipitated with Spc72-GFP and that the efficiency of their interaction decreased either when the Spc72-AA-GFP mutant was expressed instead of the wild type form of the γ-TuCR or when Cdc5 activity was inhibited using the *cdc5-as1* allele (Figures 7H, 7I, Supplementary figures 7A, 7B), in agreement with Cdc5 modulating Spc72 and Kar9 association. Similar results were obtained when Tub4-mScarlet and Spc72-GFP interaction was analyzed, indicating that Cdc5 also regulates the microtubule nucleation activity of the SPBs (Figures 7H, 7I, Supplementary figures 7A, 7B). Taken together, both the microscopy and the biochemical data clearly establish a functional link between Cdc5, Kar9 and Spc72 in the establishment of the predetermined SPB inheritance pattern in *Saccharomyces cerevisiae*.

I therefore feel that the revisions proposed here are not particularly geared to resolve the central issue of a united mechanism. I am actually disappointed at the position adopted by the authors as I was hoping that my suggestions in the original review were to the point to permit publication and overall feasible. Biochemical work on the other hand, while feasible would entail a more involved revision. The matter of Kar9 function being disrupted by Cdc5 is not lateral enough to be addressed solely by narrative, it can be easily settled – Cdc5 could be easily turned off using a MET construct – this would have overcome all the issues raised by the authors on feasibility (synthetic medium, room temperature). Also, colabelling cells for microtubules and Kar9 to document the complete range of localisations in still images following Cdc5 depletion would be a good substitute for real time microscopy if correctly performed, to meet my comments in my original review.

Following the reviewer’s alternative suggestion to her/his initial approach to demonstrate that Kar9 function is indeed disrupted by Cdc5, we have co-labelled cells for microtubules (mCherry-Tub1) and Kar9 (Kar9-sfGFP) and characterized the complete array of all possible distributions of Kar9 in still images after inactivation of Polo-like kinase activity (Figures 2E-G). In *cdc5-as1* anaphase cells expressing both Kar9sfGFP and mCherry-Tub1 protein fusions without interfering with Cdc5 activity, Kar9 was primarily observed at the daughter-oriented SPB and the (+)-end of the cytoplasmic microtubules that entered the daughter cell, in agreement with the previously described motion of Kar9 through microtubules towards the bud tip (Korinek et al., 2000; Liakopoulos et al., 2003; Maekawa et al., 2003; Yin et al., 2000). Remarkably, however, when Cdc5 activity was inhibited, Kar9 distribution was severely affected. Specifically, in the absence of the Polo-like kinase activity we observed an increased localization of Kar9 to interpolar microtubules and a more even distribution of the protein between the mother and daughter-oriented SPBs and cytoplasmic microtubule ends (Figures 2E-G). These results agree with all our previously described experimental evidences and provide further support to strongly corroborate our hypothesis that Cdc5 activity is necessary to efficiently mobilize Kar9 from the nucleus and correctly polarize this protein so that it gets asymmetrically loaded on the old SPB, hence contributing to the establishment of the predetermined SPB fate in *S. cerevisiae*.

In summary, our results indicate that a main role of Cdc5 during the establishment of SPB fate is to promote Kar9 mobilization from the nucleus and its polarization towards the daughter cell. Additionally, the control of Spc72 activity by the Polo kinase further allows to determine SPB fate and to fine-tune the distribution of the old SPB into the daughter both by regulating Spc72 and Kar9 association and also by setting up the timing at which microtubule-nucleation capacity is acquired by each SPB.